# The Musashi proteins direct post-transcriptional control of protein expression and alternate exon splicing in vertebrate photoreceptors

Fatimah Matalkah[1,6], Bohye Jeong [1,6], Macie Sheridan[2], Eric Horstick[3,4], Visvanathan Ramamurthy[1,5] & Peter Stoilov [1✉]

The Musashi proteins, MSI1 and MSI2, are conserved RNA binding proteins with a role in the maintenance and renewal of stem cells. Contrasting with this role, terminally differentiated photoreceptor cells express high levels of MSI1 and MSI2, pointing to a role for the two proteins in vision. Combined knockout of *Msi1* and *Msi2* in mature photoreceptor cells abrogated the retinal response to light and caused photoreceptor cell death. In photoreceptor cells the Musashi proteins perform distinct nuclear and cytoplasmic functions. In the nucleus, the Musashi proteins promote splicing of photoreceptor-specific alternative exons. Surprisingly, conserved photoreceptor-specific alternative exons in genes critical for vision proved to be dispensable, raising questions about the selective pressures that lead to their conservation. In the cytoplasm MSI1 and MSI2 activate protein expression. Loss of *Msi1* and *Msi2* lead to reduction in the levels of multiple proteins including proteins required for vision and photoreceptor survival. The requirement for MSI1 and MSI2 in terminally differentiated photoreceptors alongside their role in stem cells shows that, depending on cellular context, these two proteins can control processes ranging from cell proliferation to sensory perception.

[1] Department of Biochemistry and Molecular Medicine, West Virginia University, Morgantown, WV, USA. [2] Undergraduate Program in Biochemistry, West Virginia University, Morgantown, WV, USA. [3] Department of Biology, West Virginia University, Morgantown, WV, USA. [4] Department of Neuroscience, West Virginia University, Morgantown, WV, USA. [5] Department of Ophthalmology and Visual Sciences, West Virginia University, Morgantown, WV, USA. [6] These authors contributed equally: Fatimah Matalkah, Bohye Jeong. ✉email: pstoilov@hsc.wvu.edu

Mammals express approximately 500 RNA binding proteins that associate with Polymerase II transcripts to regulate pre-mRNA processing, mRNA localization, mRNA stability, and translation[1]. Networks of RNA binding proteins specific to, or highly expressed in neurons perform roles that range from diversifying the transcriptome through alternative splicing and poly-adenylation to directing transport of specific mRNA to cellular compartments for localized translation[1–4]. RNA binding proteins are essential for the development and function of the nervous system where processes such as axon guidance, synaptic plasticity, cell survival and cell excitation are tuned by their activity[4]. Many RNA binding proteins belong to families of orthologs with varying degrees of sequence homology and functional redundancy. Interestingly, even orthologs with highly similar sequences and biochemical properties are not fully redundant and can have distinct roles in the nervous system. This divergence in function can be derived from differences in expression levels across different cell types, subcellular localization, or their interactomes[5–8].

Neurons stand out among other cell types by the pervasive use of alternative exons[9,10]. The large number of alternatively spliced exons used in neurons is due to the absence of a major splicing repressor, PTBP1, and the expression of neuron specific splicing factors[4]. While the importance of neuronal splicing programs is well established through knockouts of splicing regulators, the functions of the many alternative exons are less clear. Function of individual exons is commonly assigned based on sequence conservation and the nature of the protein hosting the exon. More recently, an empirical picture of the functional impact of individual exons has started to emerge. Functions of individual exons range from fine-tuning protein interactions to regulatory switches that shut down protein expression[11–13]. Interestingly, in several cases, deletion of conserved alternative exons has failed to produce an obvious phenotype in mouse models in vivo[14–17].

Among the RNA binding proteins expressed in neurons are the Musashi family of proteins[18–20]. The founding member of the family, the *Drosophila Msi* protein was first described as a factor that maintains the undifferentiated state of stem cells by repressing the translation of the *Notch* regulator *Numb*[21]. This function of Musashi is preserved in vertebrates where its homologs, MSI1 and MSI2, are required for stem cell maintenance and are investigated for their role in cancer progression[22–25]. Subsequent studies showed that the effect of the Musashi proteins on translation is context dependent, and they can positively or negatively regulate protein translation by binding to the 3'-UTR of their target mRNAs[26–29].

We showed that in photoreceptor cells the Musashi proteins regulate alternative splicing to produce highly photoreceptor-specific isoforms of ubiquitously expressed proteins[30,31]. The Musashi proteins maintain exceptionally high protein levels in the mature retina and their expression is developmentally regulated[31]. MSI1 levels rise sharply after birth, peak between postnatal days 2 and 4, and decline afterwards. Concomitant with the decline of MSI1 protein levels, the levels of MSI2 increase and remain constant in adulthood[31]. Single deletion of *Msi1* or *Msi2* in committed rod-photoreceptor progenitor cells showed that the two paralogs are partially redundant and appear to act at time-points of retinal development that correlate with the pattern of their expression[31]. The single deletion of *Msi1* results in an early visual defect that was observed at the time of eye-opening in mice (postnatal day 16). In contrast, the removal of *Msi2* resulted in normal vision at postnatal day 16 that progressively declined with age[31]. Based on these findings, we proposed that MSI2 is involved in the maintenance of mature photoreceptor cells, while MSI1 functions in photoreceptor precursors and immature photoreceptor cells.

Using an inducible mouse model that deletes the Musashi genes in mature photoreceptors we show the Musashi proteins to be essentials for the function and viability of the photoreceptors. To our surprise, despite their reciprocal regulation during development, MSI1 and MSI2 are fully redundant in mature photoreceptors. We identified the transcripts recognized by MSI1 and MSI2 in vivo and investigated how loss of the *Msi1* and *Msi2* genes affected pre-mRNA splicing, transcript levels, and protein expression. We demonstrate that the Musashi proteins bind downstream of photoreceptor-specific exons to activate their splicing. In addition, we show that in photoreceptors the Musashi proteins act almost exclusively as post-transcriptional activators of protein expression.

## Results

**Depletion of *Msi1* and *Msi2* in mature photoreceptor cells**. We recently showed that in the retina the MSI1 and MSI2 proteins are differentially regulated and proposed that they have separate roles in development and maintenance of photoreceptor cells[31]. We tested the roles of the Musashi protein in mature photoreceptors by using tamoxifen-inducible $Cre^{ERT2}$ under the control of rod-specific *Pde6g* promoter to remove *Msi1* and *Msi2* in mature rod-photoreceptor cells[32]. Floxed ($Msi1^{flox/flox}/Msi2^{flox/flox}$) mice hemizygous for $Pde6g-Cre^{ERT2}$ were injected with tamoxifen for three consecutive days starting at postnatal day 30 to create combined *Msi1/Msi2* knockout mice. Littermates carrying the floxed alleles for *Msi1* and *Msi2* ($Msi1^{flox/flox}/Msi2^{flox/flox}$) and treated with tamoxifen were used as controls. We will refer to the $Msi1^{flox/flox}/Msi2^{flox/flox}$ treated with tamoxifen as $Msi1^{+/+}/Msi2^{+/+}$ and the mice with $Msi1^{flox/flox}/Msi2^{flox/flox}\_Pde6gCre^{ERT2}$ treated with tamoxifen as $Msi1^{-/-}/Msi2^{-/-}$. Immunocytochemistry (ICC) demonstrated that, 14 days after the first tamoxifen injection, the MSI1 and MSI2 proteins were depleted specifically from the photoreceptors (Fig. 1a). The immunofluorescence signal is lost from both the cytoplasm (inner segment, IS) and the nuclei (outer nuclear layer, ONL) of the photoreceptor cells. Consistent with the ICC data, western blot analysis showed two fold decrease in the MSI1 and MSI2 protein levels in retinal lysates from the $Msi1^{-/-}/Msi2^{-/-}$ mice (Fig. 1b and Supplementary Fig. 11).

**MSI1 and MSI2 are required for the function and survival of mature photoreceptors**. To evaluate the requirement for Musashi proteins in mature photoreceptors, we used electroretinograms (ERG) to measure the retinal response to light. We measured the dark-adapted (scotopic) and the light-adapted (photopic) responses that reflect rod and cone photoreceptor function, respectively[33]. We used repeated measures two-way ANOVA to determine the effect of the genotype and time post injection on the ERG A-Wave amplitude. We found a significant interaction of the genotype and the time after injection (Scotopic response: $F(12,1) = 19.47$, *p*-value < 0.0001; Photopic response: $F(12,1) = 10.37$, *p*-value < 0.0001). The response to light of the $Msi1^{-/-}/Msi2^{-/-}$ animals and the age-matched $Msi1^{+/+}/Msi2^{+/+}$ controls became significantly different 35 days after tamoxifen injection (Fig. 2a, b). The response to light continued to decrease rapidly thereafter and was nearly undetectable by day 105 post-injection (Fig. 2a, b).

To assess the retinal morphology following *Msi1/Msi2* deletion, we performed hematoxylin and eosin (H&E) staining on retinal cross-sections collected from $Msi1^{-/-}/Msi2^{-/-}$ and control mice between days 0 and 113 post-injection (Fig. 2c, d, and Supplementary Fig. 2). A significant effect of the genotype on the photoreceptor cell layer thickness over time was confirmed by two-way ANOVA ($F(1,259) = 61.04$, *p*-value < 0.0001). In agreement with the ERG

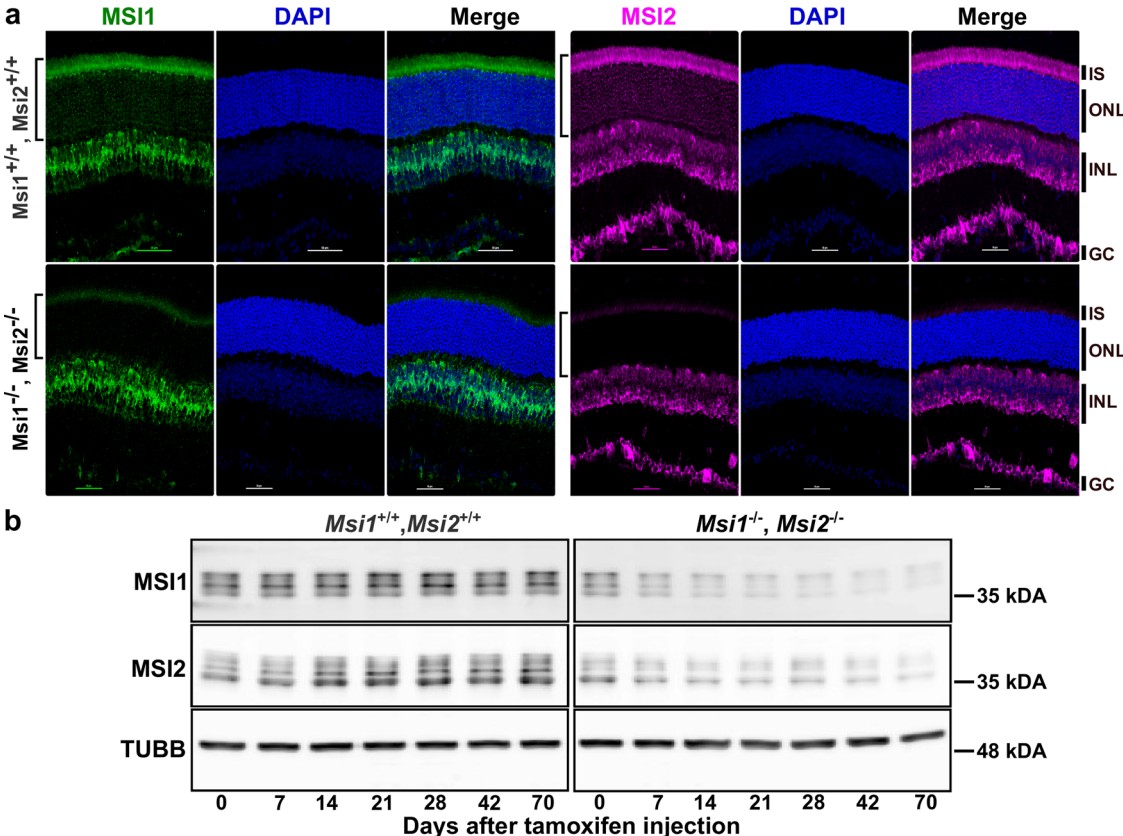

**Fig. 1 Induced double knockout of Msi1 and Msi2 in photoreceptor cells. a** Immunofluorescence micrographs of retinal cross-sections collected 14 days after tamoxifen injection at postnatal day 30 from *Msi1+/+/Msi2+/+* littermate control and *Msi1−/−/Msi2−/−*, stained for MSI1(green), MSI2 (magenta), and DAPI (blue). ONL outer nuclear layer (photoreceptor cell layer). The brackets on the left indicate the layer of photoreceptor cells from which the genes are being deleted. IS: inner segment (photoreceptor cell layer). ONL outer nuclear layer (photoreceptor cell layer), INL inner nuclear layer, GC ganglion cell layer. Objective, 40x. **b** Immunoblot of lysates from *Msi1+/+/Msi2 +/+* and *Msi1−/− /Msi2−/−* retinas collected between 0 and 70 days after tamoxifen injections at postnatal day 30 and probed with antibodies to MSI1, MSI2 and TUBB (β-tubulin, loading control). Size markers indicate the molecular weight in kDA. See Supplementary Fig. 1 for uncropped blots.

data, we did not observe significant morphological changes up to 28 days after tamoxifen injection (Fig. 2c, d and Supplementary Fig. 2). After day 28 post-injection, we observed progressive degeneration of the photoreceptor cell layer (Fig. 2d and Supplementary Fig. 2). Approximately half of the photoreceptor cells were lost by day 42 post-injection in the knockout retinas, and only one layer of photoreceptor cells remained at day 113 post-injection (Fig. 2c, d and Supplementary Fig. 2). We did not observe any significant changes in the inner retina, including the inner nuclear layer (INL) and the ganglion cell layer (GCL). Our results show that the combined deletion of *Msi1/Msi2* in mature photoreceptors leads to a rapid and progressive decline in the function and viability of photoreceptor cells that starts 4 weeks after depletion of the Musashi proteins.

**MSI1 and MSI2 are redundant in the maintenance of mature photoreceptors**. To determine if MSI2 plays a dominant role in mature retina, as the developmental regulation of MSI1 and MSI2 protein expression would suggest, we delete the *Msi1* and *Msi2* genes individually in mature photoreceptors. We confirmed the photoreceptor-specific loss of MSI1 and MSI2 protein by immunostaining retinal cross-sections obtained at day 14 post-injection (Fig. 3a, b). Western blot analysis of retinal lysates showed that tamoxifen injection required 7–14 days to ablate the proteins, in agreement with our observation of the double knockout (Fig. 3c). The MSI2 protein level was upregulated 1.4-fold in the *Msi1−/−* retina compared to the control; however, the

increase did not reach statistical significance at the number of replicates used ($n = 3$).

Neither *Msi1* nor *Msi2* single ablation had an effect on retina function (Fig. 4a). The scotopic and photopic ERG responses collected from day 7 to day 230 post injection show that the response to light of the single *Msi1* or *Msi2* knockout mice are indistinguishable from the control animals (Fig. 4a and Supplementary Fig. 4a, b). Similarly, the histology of the knockout retinas collected at day 230 post injection does not show signs of degeneration (Fig. 4b, c). These data demonstrate that Musashi proteins are fully redundant in mature photoreceptors.

**Binding of Musashi to the downstream proximal intron promotes splicing of alternative exons**. To delineate the transcripts bound by Musashi and the positions on the targets where Musashi binds, we used enhanced UV Cross-Linking and Immuno-Precipitation followed by high throughput sequencing of the associated RNA fragments (eCLIP-Seq). The MSI1 and MSI2 proteins are fully redundant in the adult retina, their RNA binding domains are 77% (RRM1) to 92% (RRM2) identical, and the two proteins recognize the same UAG binding site[34,35]. Thus, we argued that performing the eCLIP-Seq experiment on MSI1 will be sufficient to identify the targets for both proteins.

Out of 30,283 transcripts detected by RNA-Seq, 10,161 had at least one eCLIP peak enriched over input and 7849 transcripts had eCLIP peaks in the 3'-UTR (Supplementary Data 1). The eCLIP-Seq data shows that MSI1 binds predominantly to the 3'-

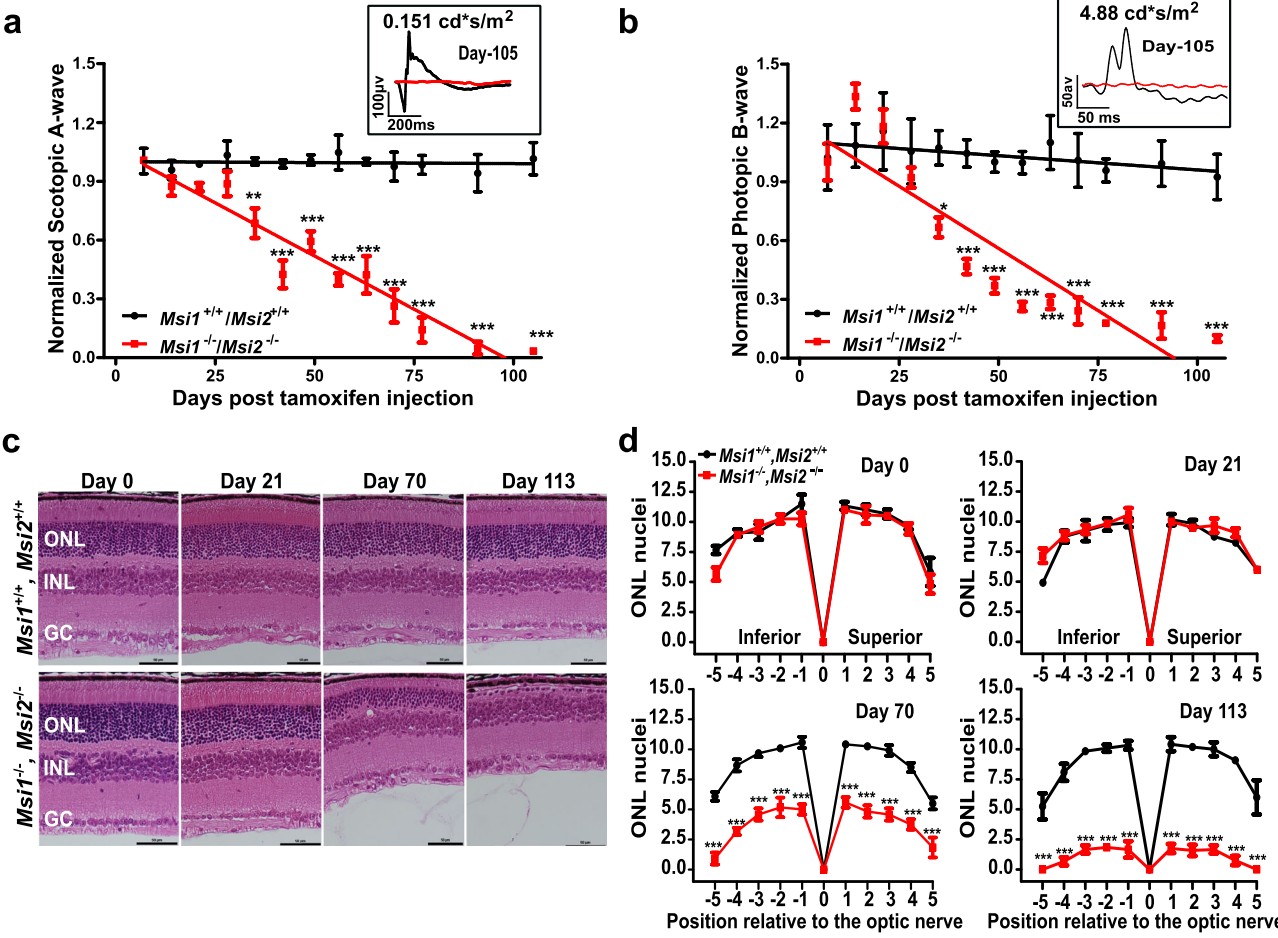

**Fig. 2 Progressive loss of response to light and retinal degeneration after double knockout of Msi1 and Msi2 in photoreceptor cells.** Time course of scotopic A-wave (**a**) and photopic B-wave (**b**) responses from $Msi1^{+/+}/Msi2^{+/+}$ (black line), and $Msi1^{-/-}/Msi2^{-/-}$ retinas (red line) following tamoxifen injection. Scotopic and photopic waveforms were obtained 0.151 cd*s/m² and 4.88 cd*s/m² flashes, respectively. Insets show representative electroretinograms from a single $Msi1^{+/+}/Msi2^{+/+}$ (black line) and $Msi1^{-/-}/Msi2^{-/-}$ (red line) mouse retina at day 105 post-tamoxifen injection. The data points from the scotopic and photopic responses are represented as mean ± SEM of 8 eyes (4 animals). Pairwise $t$-test with Bonferroni correction for multiple comparisons was used to determine the effect of genotype on the A-wave amplitude at each time point. Significance levels of the pairwise comparisons is indicated as: *$p$-value < 0.05, **$p$-value < 0.01, ***$p$-value < 0.001. **c** Outer nuclear layer degeneration in $Msi1^{-/-}/Msi2^{-/-}$ knockout retinas. Representative H&E-stained sections from the $Msi1^{+/+}/Msi2^{+/+}$ and $Msi1^{-/-}/Msi2^{-/-}$ retinas collected between day 0 and day 113 post-tamoxifen injection. ONL: outer nuclear layer (Photoreceptor nuclei), INL: inner nuclear layer, GC: ganglion cells. 40X objectives and scale bar represents 50 μm. **d** Spider plots displaying the thickness of the ONL as the number of nuclei measured at ten points stepped by 400 μm from the optical nerve at different time points post-tamoxifen injection. Data are shown as mean ± SEM. Pairwise $t$-test with Bonferroni correction for multiple comparisons was used to determine the effect of the genotype on the outer nuclear layer thickness at each time point. Significance levels of the pairwise comparisons is indicated as: *$p$-value < 0.05, **$p$-value < 0.01, ***$p$-value < 0.001. Supplementary Data 15 contains the source data underlying the graphs.

UTRs of mRNA (59.7% of binding sites, Fig. 5a, c) and to introns of pre-mRNA (32.7% of binding sites, Fig. 5a, e). Motif enrichment analysis of the sequence surrounding the eCLIP crosslink sites by HOMER and MEME software suites identified as enriched motifs centered on a UAG core (Supplementary Fig. 5). This result is in agreement with the UAG binding site sequence for the Musashi proteins derived from in vitro binding and structural studies[34–36]. The crosslink frequency peaks at position -1 relative to the top motif identified by DREME, BUAG, indicated direct binding of MSI1 (Fig. 5b).

Previously we identified a photoreceptor-specific alternative splicing program by comparing the splicing in wild-type retina to that in retina that is devoid of photoreceptor cells due to knockout of the Aipl1 gene[30]. Motif enrichment analysis suggested a role for the Musashi proteins in controlling this program and we demonstrated that the splicing of at least one exon, exon 2 A in the *Ttc8* gene, is activated by MSI1 bound to the downstream

intron[30]. Here we sought to determine on a global scale how the Musashi proteins are regulating alternative splicing in photoreceptor cells in vivo. Analysis by RNA-Seq of alternative splicing in $Msi1^{-/-}/Msi2^{-/-}$ retina 21 days after tamoxifen injection identified 165 exons that had reduced inclusion levels and 115 exons that were upregulated in the knockout (Supplementary Data 2, 3, and 4). Out of the 165 exons downregulated in the Msi1/Msi2 knockout, 52 were also significantly downregulated in the photoreceptor-devoid retina of the Aipl1 knockout mice, with another 40 exons showing the same direction of change but not reaching statistical significance in the Aipl1 knockout retina (Supplementary Data 3)[30]. None of the significantly downregulated exons in the Msi1/Msi2 knockout retina were significantly upregulated in the Aipl1 knockout (Supplementary Data 4). We did not observe a correlation between the exons significantly upregulated in the $Msi1^{-/-}/Msi2^{-/-}$ retina and the exons differentially spliced in the Aipl1 knockout retina.

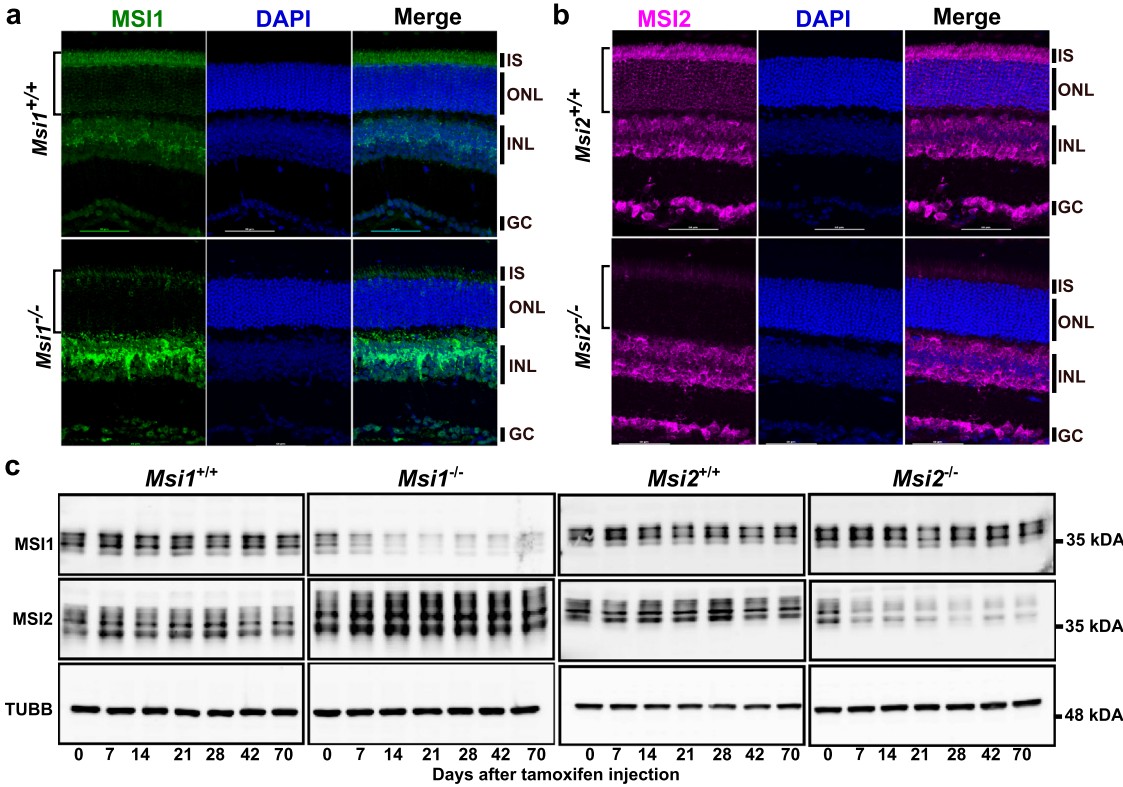

**Fig. 3 Induced single knockouts of Msi1 or Msi2 in photoreceptor cells.** Immunofluorescence micrographs of retinal cross-sections collected 14 days after tamoxifen injection at postnatal day 30 from *Msi1*−/− mice and *Msi1*+/+ littermates (**a**) or *Msi2*−/− mice and *Msi2*+/+ littermates (**b**). Sections were stained with antibodies to MSI1 (green), MSI2 (magenta) and DAPI (blue). The brackets on the left indicate the layer of photoreceptor cells from which the genes are being deleted. IS inner segment (photoreceptor cell layer), ONL outer nuclear layer (photoreceptor cell layer), INL inner nuclear layer, GC ganglion cell layer. Objective, 40X. **c** Immunoblot of lysates from *Msi1*+/+ and *Msi1*−/− retinas collected between 0 and 70 days after tamoxifen injections and probed with antibodies to MSI1, MSI2 and TUBB (loading control). Size markers indicate the molecular weight in kDA. See Supplementary Fig. 3 for uncropped blots.

Our previous work suggested that the Musashi proteins promote inclusion of alternative exons by binding downstream of the exon in the adjacent intron. To determine if this mode of regulation is common in vivo we combined the eCLIP-Seq and RNA-Seq data to build an RNA splicing map of a meta cassette exon (Fig. 5d). The splicing map shows significant enrichment of Musashi protein binding to the downstream introns proximal to the exons upregulated by the Musashi proteins (exons down-regulated in the knockout). No significant enrichment of Musashi binding sites was observed for exons repressed in the wild-type animals compared to the Musashi knockouts.

**Photoreceptor-specific microexons in the *Ttc8*, *Cep290*, *Cc2d2a*, *Cacna2d4* and *Slc17a7* genes are dispensable.** Considering the requirement of Musashi proteins for vision and their role in promoting splicing of photoreceptor-specific exons, we next tested if photoreceptor-specific alternative splicing variants are required for vision. Using CRISPR/Cas9 we deleted photoreceptor-specific exons in the *Ttc8*, *Cep290*, *Cc2d2a*, *Cacna2d4* and *Slc17a7* genes[30]. The exons in *Ttc8*, *Cep290*, *Cc2d2a* and *Cacna2d4* genes are microexons, 30 nt or less in length, showing sequence conservation across vertebrates that is traceable to fish (Supplementary Fig. 6a). We further confirmed by RT-PCR that the four alternative exons are used in zebrafish and are included at high rate in the zebrafish eye (Supplementary Fig. 6b). The photoreceptor-specific exon in *Slc17a7* is confined to rodents and serves as a control representing an evolutionary novel exon that is less likely to impact the function of the host protein. The exons in the *Ttc8*, *Cep290*, *Cc2d2a* and *Slc17a7* genes were

downregulated in our *Msi1*/*Msi2* double knockout mice. Deletion of each exon was confirmed by sequencing the alleles after the founders have been outcrossed (Supplementary Fig. 8). RT-PCR from retinal samples showed the expected expression of exon skipped isoform (Fig. 6a).

We examined the visual function of the exon knockout mice by ERG between one and 12 months of age. We did not observe significant differences in the response to light of the exon knockout mice compared to wild-type controls (Fig. 6b). Similarly, H&E stained retinal sections from the exon knockout mice had normal morphology (Fig. 6c). The phenotypes of the individual exon knockout mice may have been too subtle to detect on their own. Thus, we crossed the *Ttc8*, *Cep290*, and *Cc2d2a* exon knockouts to create a homozygous triple exon knockout mouse line. As all three proteins are part of the primary cilium and are critical for cilium biogenesis, we expected the individual exon knockout phenotypes to be amplified in the combined knockout. As with the single exon knockout mice we did not observe changes in the function or morphology of the retina of the triple exon knockout animals (Fig. 6b, c).

**MSI1 and MSI2 are post-transcriptional activators of protein expression in photoreceptor cells.** Consistent with the previously described role of the Musashi proteins in regulating mRNA translation, our CLIP-seq data showed pervasive binding of MSI1 to 3′-UTRs (Fig. 5a, c). To determine the effect of Musashi on the protein expression, we analyzed the changes in mRNA and protein levels in *Msi1*−/−/*Msi2*−/− retina where both MSI1 and MSI2 were depleted from mature photoreceptor cells. For RNA-

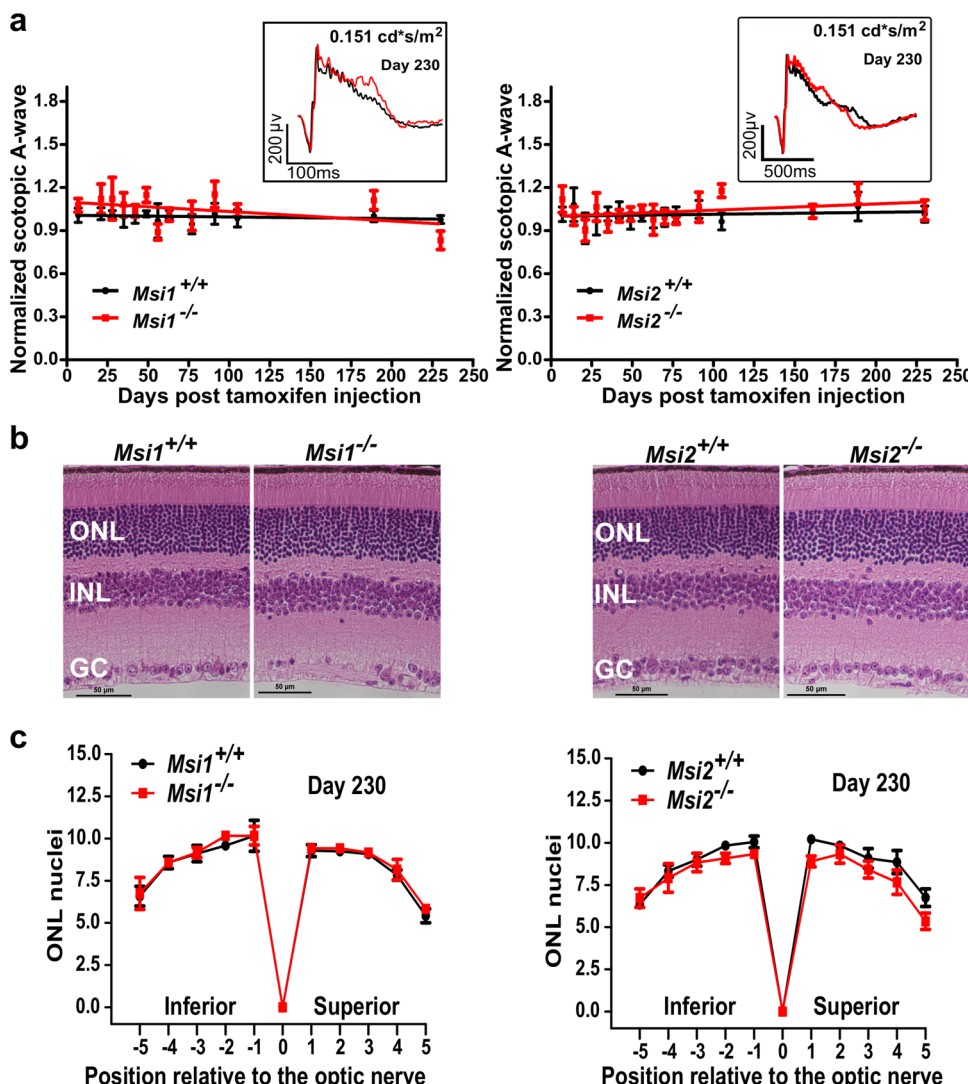

**Fig. 4 Normal photoreceptor response to light and retinal morphology in the single knockouts Msi1 or Msi2 in photoreceptor cells. a** Scotopic mean A-wave response from single *Msi1* −/− (left) or *Msi2* −/− (right) knockouts in photoreceptors (red lines) and littermate controls (black lines) recorded between 0 and 230 days post-tamoxifen injection. Scotopic waveforms were obtained using 0.151 cd-s/m² flashes. The insets of panels A and B show representative scotopic (dark-adapted) electroretinograms from a single knockout (red) or control (black) retina at D230 post-injection using 0.151 cd-s/m² flashes. Scotopic waveforms were obtained after overnight dark adaptation using 0.151 cd-s/m² flashes. **b** Representative H&E-stained sections from retinas of single *Msi1* −/− (left) or *Msi2* −/− (right) knockouts in photoreceptor cells and littermate controls collected 230 days post-tamoxifen injection. ONL: outer nuclear layer (Photoreceptor nuclei), INL: inner nuclear layer, GC: ganglion cells. 40X objectives, scale bar, represents 10 μm. **c** Spider plots of ONL thickness for single *Msi1* −/− (left) or *Msi2* −/− (right) knockouts (red) and littermate controls (black). The plots display the thickness of the ONL as the number of nuclei measured at ten points stepped by 400 μm from the optical nerve in retinas collected 230 days after tamoxifen injection. Data are shown as mean ± SEM of 8 eyes, from four animals. Supplementary Data 15 contains the source data underlying the graphs.

Seq and quantitative proteomics, we used retinas that were collected 21 days after tamoxifen injection. At this time the Musashi proteins were depleted from photoreceptor cells, while the knockout retina had normal morphology and response to light (Fig. 2). Thus, the effect of mRNA and protein expression could be analyzed without the confounding effects of photoreceptor cell death.

We used isobaric labeling and mass spectroscopy to compare the expression of 8021 proteins in knockout and control retina. Of these proteins 165 showed significant differences in expression (at least 1.5 fold change in protein levels with adjusted p-value at or below 0.01) between the control and knockout retina. Of the proteins with significant changes 98 had reduced expression and 67 had elevated expression in the knockout retina (Supplementary Data 5). As expected, MSI1 and MSI2 protein levels were

decreased by >2-fold in the retina of the *Msi1/Msi2* knockout mice (Supplementary Data 5 and Supplementary Fig. 10), consistent with the change in expression observed by western blot (Fig. 1, and Supplementary Fig. 11). Gene Ontology and KEGG pathway enrichment analysis showed that proteins downregulated in the knockout retina were strongly associated with categories related to phototransduction, photoreceptor cell structure, and photoreceptor homeostasis (Fig. 7, Supplementary Fig. 11 and Supplementary Data 6). The reduced expression of proteins in these categories is not a consequence of photoreceptor cell death or degeneration for two reasons. First, morphologically and physiologically, the retinas of the knockout animals are normal at this stage. More importantly, the levels of multiple proteins that are specific to photoreceptors or are abundantly expressed in photoreceptor cells were unchanged or even

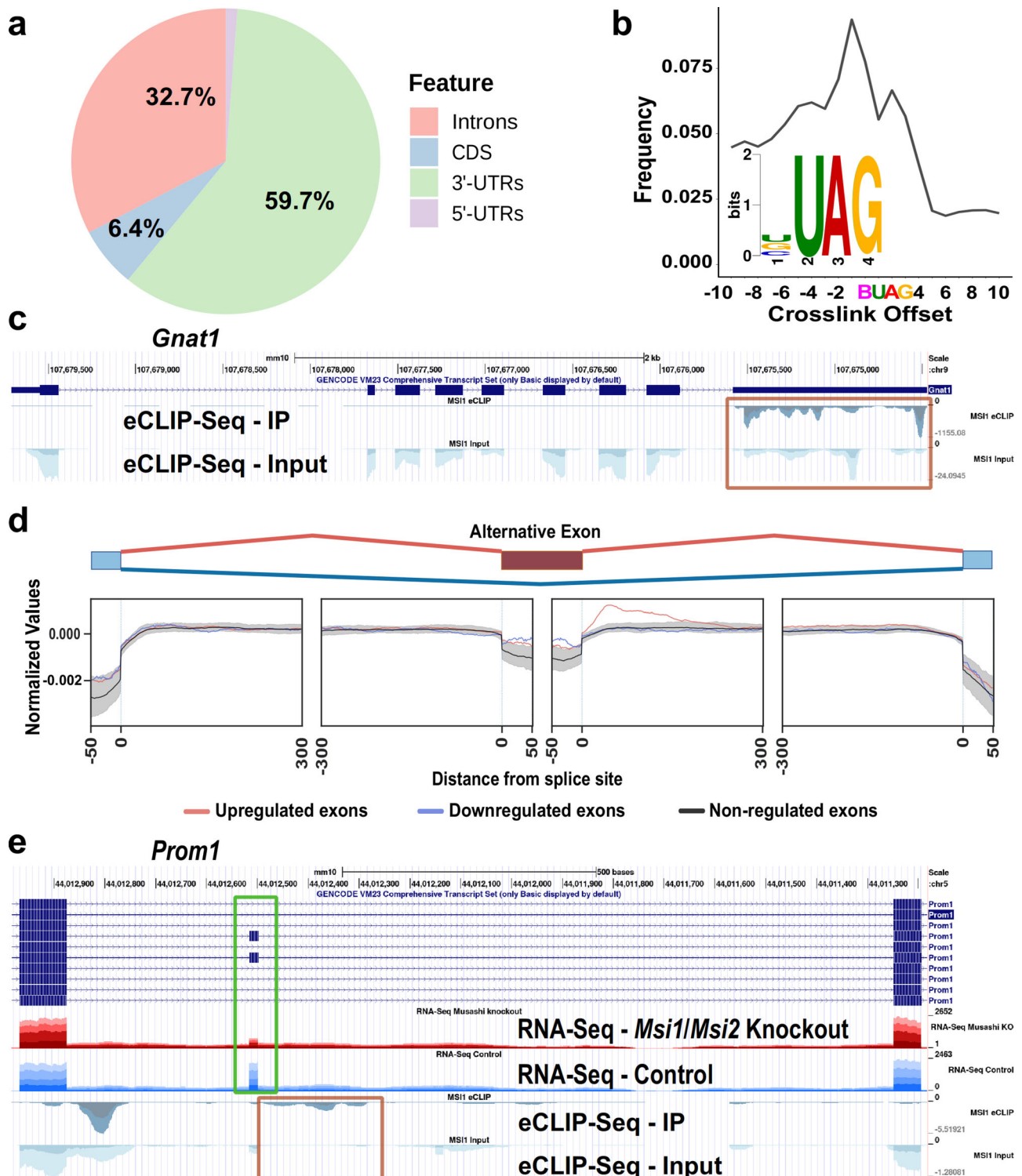

**Fig. 5 In the retina MSI1 binds to UAG motifs located predominantly in introns and 3'-UTRs. a** Distribution of MSI1 binding sites as identified by eCLIP-Seq on mouse retinal samples across mRNA features. **b** eCLIP crosslink frequency relative to the top scoring motif (BUAG) identified by DREME. **c** UCSC Genome browser tracks showing the eCLIP-seq signal enrichment over the 3'-UTR of the *Gnat1* gene (orange box). Replicates are stacked and indicated by different shading colors. Scales are 0 to −1155 for the eCLIP IP and 0 to −24 for the eCLIP input. **d** RNA binding protein map showing MSI1 binding relative to an alternative metaexon. Exons upregulated by the Musashi proteins (downregulated in the *Msi1/Msi2* double knockout) are shown in red, exons downregulated by the Musashi proteins are shown in blue and alternative exons remaining unchanged in the knockout are shown in black. Gray shading indicates the 99.5% confidence interval derived from 1000 random permutations. MSI1 binding sites are enriched downstream of alternative exons upregulated by the Musashi proteins. **e** UCSC Genome browser tracks showing MSI1 binding (orange box) downstream of an alternative exon in the *Prom1* gene regulated by the Musashi proteins (green box). RNA-Seq tracks show the read density for retinal samples derived from photoreceptor-specific *Msi1/Msi2* double knockouts and matched controls. Replicates are stacked and indicated by different shading colors. Scales are 0 to −5.5 for the eCLIP IP and 0 to −1.3 for the eCLIP input.

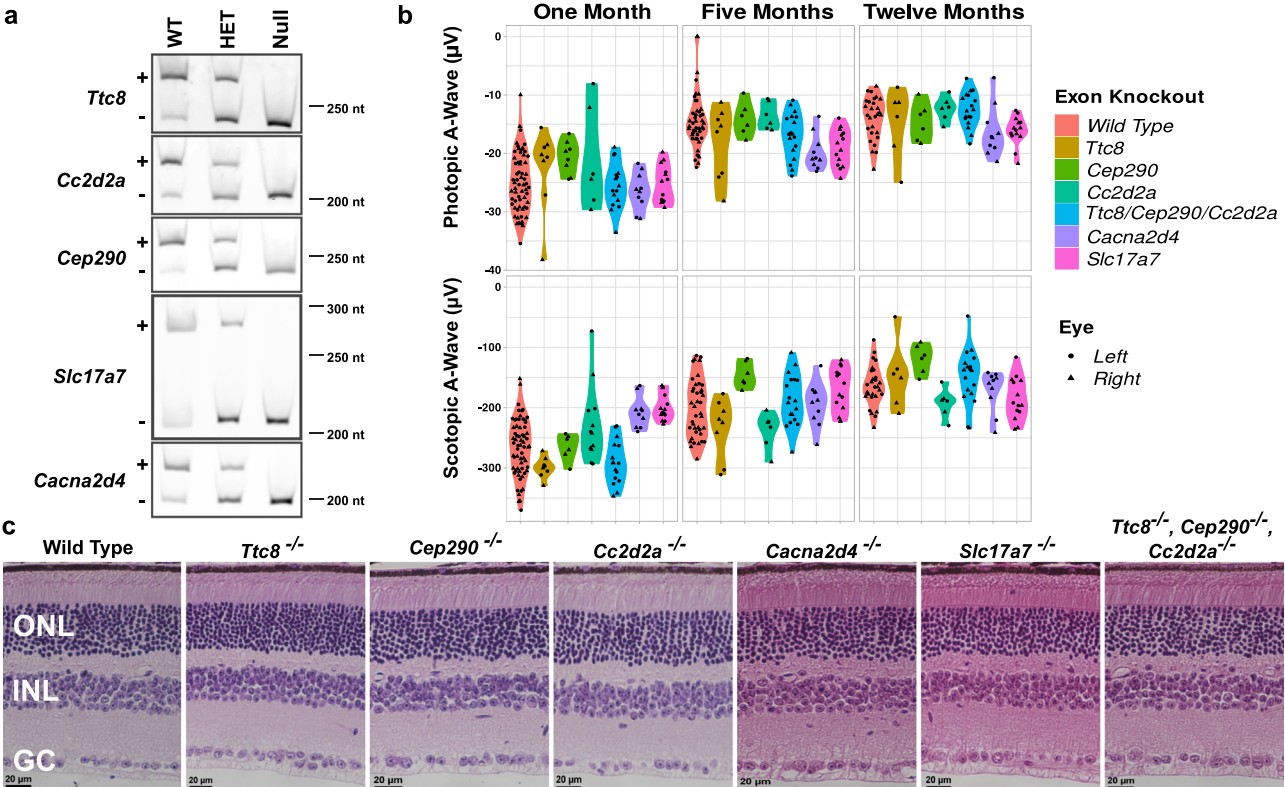

**Fig. 6 Normal photoreceptor response to light and retinal morphology of knockouts of photoreceptor-specific exons in the Ttc8, Cc2d2a, Cep290, Cacna2d4, and Slc17a7 genes. a** RT-PCR of retinal samples showing loss of the photoreceptor-specific mRNA isoforms in the exon knockout animals. RNA was extracted from the retinas of wild-type animals (WT), and heterozygous (Het) and homozygous (Null) exon knockouts. Isoforms including and skipping the alternative exon are indicated by "+" and "−", respectively. Size markers indicate the fragment size in nucleotides. Uncropped gel images are shown on Supplementary Fig. 7. **b** Violin plots of the photopic and scotopic A-wave intensities at postnatal days 30, 150 and 356 collected from the single and triple (Ttc8, Cc2d2a, Cep290) exon knockouts. Supplementary Data 16 contains the source data underlying the graphs on **b**. **c** Representative H&E-stained retinal sections from wild-type mouse, knockouts of photoreceptor-specific microexons in the Ttc8, Cep290, Cc2d2a, Canca2d4 and Slc17a7 genes, and combined deletion of the microexons in the Ttc8, Cep290, Cc2d2a genes. The samples were collected at 12 months of age. ONL outer nuclear layer (Photoreceptor nuclei), INL inner nuclear layer, GC ganglion cells. 40X objectives, scale bar, represents 20 μm.

increased (Fig. 7). Examples include proteins with functions in phototransduction (RCVRN), outer segment structure (PRPH2, PROM1), primary cilium structure (CC2D2A, CEP290), intra-flagellar transport (IFT80, IFT140), ion transport (ATP1B2), and protein transport (RD3).

In contrast to the downregulated proteins, most of the proteins with increased expression in the knockout retina were associated with Gene Ontology terms and KEGG pathways involved in cell proliferation, extracellular matrix structure, immune response, and angiogenesis (Supplementary Data 6). Closer examination of the upregulated proteins revealed proteins (GFAP, CLU, STAT3, JUNB, IRF9, A2M, B2M, complement components) that are expressed at elevated levels across various models of retinal degeneration[37–40]. Single cell RNA-Seq of the Cwc27[fs] model of retinal degeneration indicated that many of the upregulated genes are expressed by glia[37]. To determine how MSI1 and MSI2 regulate protein expression in photoreceptor cells, we defined two sets of genes. The first set were genes that are either specifically expressed in photoreceptor cells or have at least two fold higher expression in photoreceptors compared to any other cell type in the retina. The second set contained genes that are either not expressed in photoreceptor cells or have at least two fold lower expression compared to other retinal cell types. The two sets of genes were derived from differential expression analysis of single cell RNA-Sequencing data by Macosco et al.[41] We will use "photoreceptor-specific" as a shorthand for the subset of genes highly expressed specifically in photoreceptor cells, with the

understanding that many of these genes are also expressed in other cell types, albeit at lower levels. Most proteins with significantly lower expression in the $Msi1^{-/-}/Msi2^{-/-}$ retina and without a significant change in their mRNA levels, belonged to the photoreceptor-specific set of genes (Fig. 8a and Supplementary Data 8). Only two photoreceptor-specific proteins, PROM1 and IMPG2, had markedly higher expression in the knockouts (Fig. 8a). All 39 proteins with altered expression derived from "photoreceptor-specific" genes (blue rombs on Fig. 8a) contained MSI1 eCLIP peaks in their 3'-UTRs. A cumulative plot of the changes in protein and RNA expression from the photoreceptor-specific genes shows a global trend in reduced protein levels that were not matched by a corresponding decrease in transcript levels (Fig. 8b). Taken together our data demonstrates that the Musashi proteins promote protein expression post-transcriptionally.

**MSI1 promotes translation of recombinant Gnat1.** To demonstrate a direct activation of protein expression by Musashi we examined the effect of MSI1 on protein expression from Gnat1 clones carrying full length 3'-UTR in a heterologous system, NIH 3T3 cells. The NIH 3T3 cells were chosen for the low levels of endogenous MSI1 and MSI2 protein expression. We created Gnat1 clones that contained either wild type 3'-UTR or a mutant 3'-UTR in which the TAG sites were changed to TGA to prevent Musashi binding. The wild type and mutant clones carried HA and T7 epitope tags, respectively. The clones were mixed

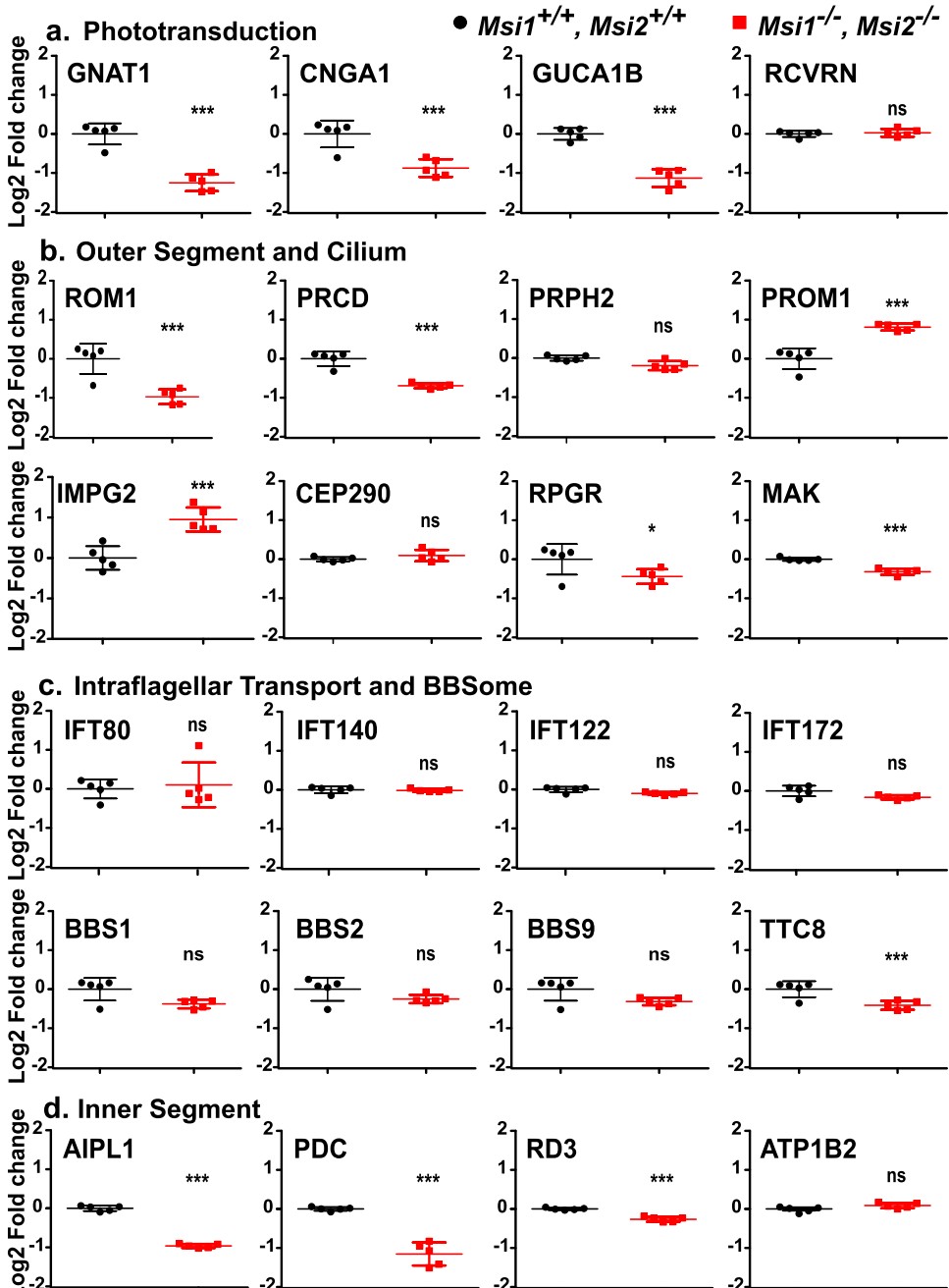

**Fig. 7 Expression of proteins critical for photoreceptor function after induced knockout of Msi1 and Msi2 in photoreceptor cells.** Boxplots representing the log2 of the fold change in protein expression relative to the median of the control. Retinal samples were collected 21 days after inducing the knockout at postnatal day 30. Protein levels were determined using isobaric labeling and mass-spectroscopy. Boxplots represent a selected set of proteins that are components of the phototransduction pathway (**a**), outer segment primary cilium structure (**b**), intraflagellar transport complex and BBSome (**c**), and the inner segment (**d**). The data is represented as mean ± SEM of five replicates. Significance level is indicated as: *adjusted *p*-value < 0.05, **adjusted *p*-value < 0.01, ***adjusted *p*-value < 0.001. Supplementary Data 18 contains the source data underlying the graphs.

together in equal amounts and co-transfected in NIH 3T3 cells with vectors expressing either GFP, *Pcbp2*, or *Msi1*. PCBP2 is a RNA binding protein that like MSI1 and MSI2 is abundantly expressed in photoreceptors, shuttles between the nucleus and the cytoplasm, and regulates splicing and translation[42,43]. The products of the wild type and mutant clones were distinguished by the HA (wild type) and T7 (mutant) epitope tags (Fig. 8c, d). The epitope tags were detected by antibodies on Western blot or by hydrolysable probes in multiplexed RT-qPCR, to measure protein and mRNA expression respectively. The effect of the co-

transfected constructus on *Gnat1* expression was measured as the change in the ratio of the HA to the T7 signal.

One-way ANOVA found a significant effect of the co-transfected expression vector on the HA/T7 GNAT1 protein ratio (F(2,15) = 50.85, *p*-value = 2.08*10-7). Tukey HSD post-hoc test showed that MSI1 has a significant (*p*-value < 1*10-7) effect on the GNAT1 protein expression compared to the vectors expressing GFP or PCBP2 (Fig. 8c, d). The analysis of the *Gnat1* mRNA levels also revealed a significant effect of the co-transfected expression construct (one-way ANOVA F(2,15) = 8.19, *p*-value = 0.004).

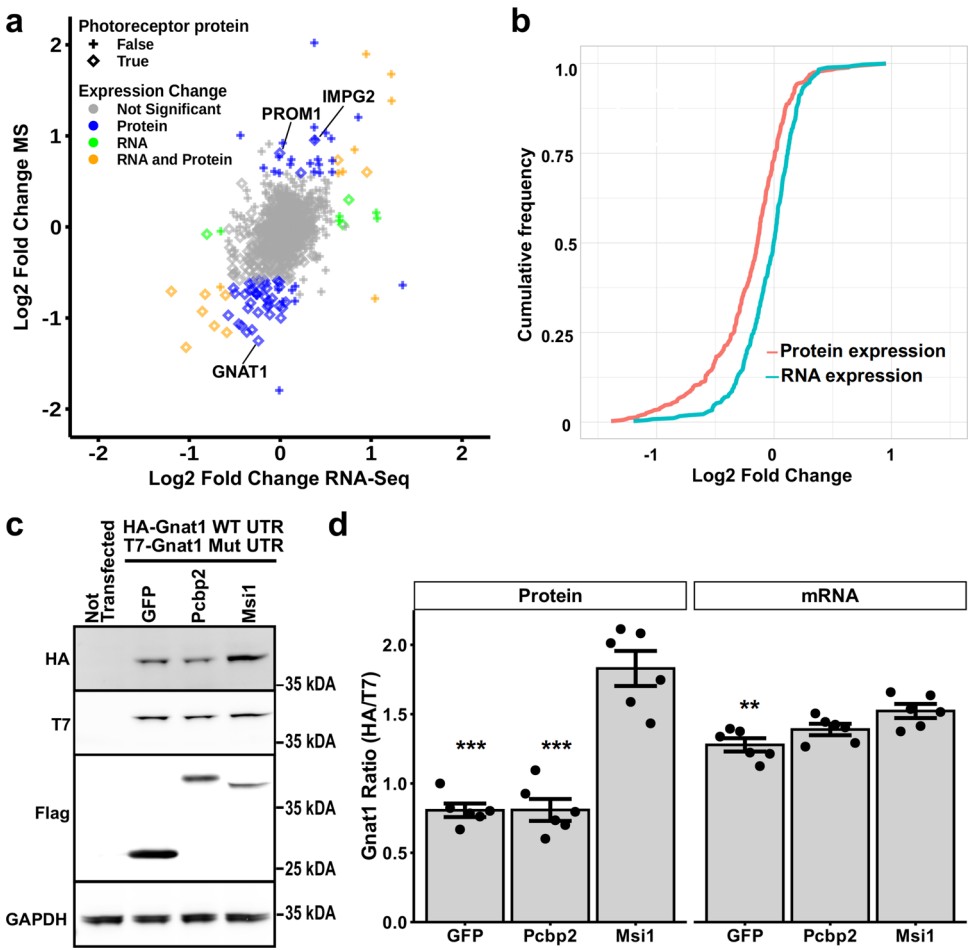

**Fig. 8 In photoreceptor cells MSI1 and MSI2 act to promote protein expression post-transcriptionally. a** Scatter plot comparing protein and transcript levels changes after double *Msi1/Msi2* knockout in mature photoreceptor cells. Protein expression was quantified by isobaric labeling and mass spectrometry. Transcript expression levels were determined by RNA-Seq. Genes that are highly expressed specifically in photoreceptor cells are shown as rombs and genes highly expressed outside of photoreceptor cells are shown as crosses. Color indicates significant changes in protein levels alone (blue), in mRNA levels alone (green), in both protein and mRNA levels (orange), or no significant change in expression (gray). **b** Cumulative frequency plots showing the deletion of *Msi1/Msi2* in mature photoreceptors leads to broad decrease of photoreceptor protein levels in excess to changes in transcript levels (Kolmogorov–Smirnov *p*-value = 1.1*10$^{-12}$). **c** Western blot analysis of recombinant GNAT1 expression in NIH 3T3 cells. Cells were transfected with equal amounts HA-tagged *Gnat1* clone with wild type 3'-UTR and T7-tagged *Gnat1* clone with mutant 3'-UTR that lacks Musashi binding sites. Each transfection included one of the following: vector expressing flag-tagged GFP, vector expressing flag-tagged *Pcbp2*, and vector expressing flag-tagged *Msi1*. Non transfected cells were included as a control for the specificity of the antibodies. TUBB serves as loading control. Size markers indicate the molecular weight in kDA. **d** Ratios of the HA (wild type) to T7 (mutant) tagged GNAT1 proteins and ratios of the corresponding RNA transcripts in the transfected NIH 3T3 cells. The data is represented as mean ± SEM of six replicates. The statistical significance of the effect of MSI1 in pairwise comparisons to the controls was determined by Tukey HSD. Significance level is indicated as: *\*p*-value < 0.05, **\*\*p*-value < 0.01, ***\*\*\*p*-value < 0.001. Supplementary Data 15 contains the source data underlying the graph on **d**. Uncropped blots used underlying the data shown on **c**, **d** are shown on Supplementary Fig. 13.

The observed effect was due to a marginal increase in *Gnat1* transcript levels in response to MSI1 compared to GFP (Fig. 8d). The two fold increase in GNAT1 protein expression without a corresponding increase in transcript levels recapitulates the regulation of GNAT1 by Musashi that we observed in photo-receptor cells and points to a role for the Musashi proteins as activators of translation.

## Discussion

Here we demonstrate that MSI1 and MSI2 are critical for pho-toreceptor function and survival. Disruption of the two genes resulted in rapid loss of vision and retinal degeneration (Fig. 2 and Supplementary Fig. 2). The Musashi proteins were fully redundant in mature photoreceptors and the single *Msi1* or *Msi2* knockouts did not have detectable phenotypes. It is unclear why *Msi1* and *Msi2* were only partially redundant during retinal

development while they were fully redundant in mature photoreceptor cells.

While hundreds of mutations in dozens of genetic loci lead to loss of vision, to date no vision defects have been associated with mutations in the *Msi1* and *Msi2* genes despite their critical function in photoreceptor cells. The lack of disease causing mutations associated with the Musashi genes is likely a combi-nation of their redundancy and their critical role in stem cell maintenance. The redundancy of the two proteins ensures that mutations in one of the genes will be complemented by the other. At the same time, loss of both *Msi1* and *Msi2* will result in embryonic lethality and preempt the observation of retinal phenotypes.

Photoreceptor cells express a distinct splicing program that utilizes a large number of microexons. Motif enrichment analysis suggested that the inclusion of photoreceptor-specific exons is

driven by the Musashi proteins[30]. Recent studies on mouse tissues, flow sorted retinal cells, and human retinal organoids showed a similar photoreceptor-specific splicing program directed by Musashi[42,44]. Here we demonstrate that in photoreceptors the Musashi proteins directly promote the splicing of alternative exons by binding to the downstream intron. The Musashi proteins promoted the inclusion of more than half of the exons we previously defined as photoreceptor specific[30]. This leaves a sizable population of photoreceptor-specific exons that rely on other factors for their splicing. Recent studies have highlighted two such factors, PCBP2 and SRRM3, that can either act independently or cooperate with Musashi to promote splicing of alternative exons in photoreceptors[42,45].

In addition to the exons activated by the Musashi proteins, a comparable number of exons appeared to be repressed by them. The repressed exons lack enrichment of MSI1 binding to them or to the adjacent introns when compared to alternative exons that are not regulated by Musashi. It is possible that the Musashi proteins have more than one mode of directly repressing splicing and our dataset does not have sufficient power to detect these interactions as enriched. It is also likely that many of the repressed exons are not direct targets of the Musashi proteins but are regulated by factors whose expression is controlled by MSI1 and MSI2. For example, the Musashi proteins negatively regulate the expression of SRSF9 (Supplementary Data 5). Consequently, the splicing of exons dependent on SRSF9 will be repressed indirectly by the expression of the Musashi proteins in photoreceptors.

Use of alternative microexons is a hallmark of the nervous system. The majority of these alternative exons are conserved which has led to the conclusion that they are essential in neurons. For several microexons such essential function has been demonstrated using animal knockout models[12,46–48]. We reasoned that exons that are conserved in vertebrates and are specifically used in photoreceptor cells will likely have an important role in vision. We deleted four such conserved exons in the *Ttc8*, *Cep290*, *Cc2d2a* and *Cacna2d4* genes, and one exon in the *Slc17a7* gene that is present only in rodents and is potentially nonessential (Supplementary Fig. 8). All five exons are photoreceptor-specific and are included in nearly all of the transcripts from the corresponding genes in photoreceptor cells[30,42]. The genes hosting four of the exons, *Ttc8*, *Cep290*, *Cc2d2a* and *Cacna2d4*, are essential for vision[49–52]. To our surprise the exon knockout animals did not show a detectable phenotype. A triple knockout of the exons in *Ttc8*, *Cep290*, and *Cc2d2a*, all essential components of the cilium, also lacked an adverse phenotype. The absence of phenotype in our exon knockout animals raises questions about the nature of the selective pressures that have led to the conservation of these exons. It is possible that the selective pressures are exerted by factors that are absent from the environment under which laboratory mice are reared. An alternative explanation is that these exons do not have function and do not alter the properties of the proteins. The conservation of such functionally neutral exons will be due to purifying selection that eliminates mutations negatively affecting the function of the host protein and tolerates wholesale deletion of the exon. In support of this model, the photoreceptor-specific exons in *Ttc8* and *Cc2d2a* are absent from several species (Supplementary Fig. 6). Regardless of the conservation mechanism, our results caution against using sequence conservation as a sole indicator of function.

The effect of the Musashi proteins on protein translation is context dependent. In flies and cultured mammalian cell lines the Musashi proteins repress translation of *Numb* and p21$^{Waf1/Cip1}$, while in frog oocytes they activate early translation of Mos and Cyclin B5 after progesterone stimulation[21,26,53,54]. Furthermore, recent genome wide studies integrating ribosome profiling and RNA binding data showed that within the same cell MSI1 and MSI2 repress translation of certain transcripts while activating others[55,56]. The transcriptome-wide studies also show that translation of relatively few of the large numbers of transcripts that are bound by the Musashi proteins is affected when the levels of MSI1 and MSI2 are manipulated[56].

Here we present an integrative analysis of the effect of MSI1 and MSI2 on protein translation in photoreceptor cells. To isolate the signal derived from photoreceptor cells, we focused our analysis on transcripts highly expressed in photoreceptors compared to other retinal cells and relied on the fact that photoreceptor cells are the dominant cell type in the retina comprising approximately half of the cells in that tissue. As we are excluding from our analysis transcripts that are expressed in other cells of the retina at levels comparable to those in photoreceptors, we derived a broad but far from comprehensive picture of the effect of MSI1 and MSI2 on the transcriptome and proteome of photoreceptor cells. The amount of material required for RNASeq, mass-spectrometry, and eCLIP-Seq experiments did not allow us to perform the experiments in parallel on samples from the same animal. Samples from different animals were used for each experiment. While there is an excellent correlation across replicates, it should be noted that changes in protein and RNA expression were not determined within the same animal.

We demonstrate that the combined deletion of *Msi1* and *Msi2* alters protein expression from a set of transcripts without significantly affecting the levels of these transcripts. In agreement with the previously published transcriptome-wide studies, our integrative analysis shows that the levels of relatively few proteins are affected in the Musashi knockout mice, compared to the thousands of transcripts bound by MSI1 at their 3'-UTRs.

In photoreceptors, the Musashi proteins act largely to promote protein expression. Furthermore, the translation of at least one of the targets identified in this work, *Gnat1*, is directly stimulated by MSI1 in a heterologous system. We observed only two cases, PROM1 and IMPG2, where the combined knockout of *Msi1* and *Msi2* resulted in elevated protein levels. This is an unexpected finding in light of the canonical view of the Musashi proteins as repressors of translation. As the Musashi proteins are regulating both pre-mRNA splicing in the nucleus and protein expression in the cytoplasm there are some genes affected by both modes of regulation, e.g. *Prom1*, raising the question of potential coordination between pre-mRNA splicing and protein expression. We do not see evidence for such coordination in our data.

The proteins regulated by MSI1 and MSI2 are central to the function and survival of photoreceptor cells. They include the products of a number of genes that are commonly mutated in blinding disease (GNAT1, CNGA1, PRCD, ROM1, AIPL1, PDE6A, etc). Photoreceptor cells need to produce high levels of these proteins in order to replace their outer segments every 10 days. This renewal process does not reuse proteins already present in the outer segment. Instead, new membranes and proteins are delivered to the bottom of the outer segment stack, while old segments are phagocytosed and digested by retinal pigmented epithelium from the top of the stack. Reduced rate of production of outer segment proteins and the chaperones that fold them will impede the outer segment renewal process leading to loss of vision and degeneration of the photoreceptor cells.

## Methods

**Animals**. All animal experiments were conducted with the approval of the Institutional Animal Care and Use Committee at West Virginia University. Both males and females were used in all experiments. The mouse lines in this study were in the C57BL6/J background and devoid of the naturally occurring *rd1* and *rd8* alleles. The mice were genotyped at weaning unless otherwise specified in the results section. The primers used for genotyping of the targeted alleles are listed in Supplementary Data 9.

Mice carrying *Msi1* *flox/flox* and *Msi2* *flox/flox* were provided by Dr. Christopher Lengner from the University of Pennsylvania (Li et al., 2015; Park et al., 2014). Mice carrying the floxed alleles were crossed with *Pde6g-Cre*ERT2 mice to enable photoreceptor-specific conditional knockout of *Msi1* and *Msi2* (Koch et al., 2015). The conditional deletion of *Msi1* and *Msi2* in mature photoreceptor cells was induced by intraperitoneal injection of tamoxifen in corn oil (Sigma-Aldrich catalog #T5648-1G) at concentrations of 100 mg/kg body weight for three consecutive days.

The knockouts of the photoreceptor-specific exons in *Ttc8*, *Cep 290*, and *Cc2d2a* were created using CRISPR/Cas9. Two guide RNAs targeted at sites upstream and downstream of each alternative exon were used to cause full deletion of the exon and the proximal parts of the introns. The guide RNAs were synthesized by Synthego and IDT. The guide RNA targeting sequences are listed in Supplementary Data 10. The guide RNAs and Cas9 (Thermo Fisher) were assembled into ribonucleoprotein complexes and electroporated into zygotes by the WVU transgenic core facility. The founders were back-crossed to C57BL6/J mice (Jackson Laboratory) for five generations. To map the borders of the deletions, the exon knockout alleles were amplified by PCR using the genotyping primers and sequenced by Sanger sequencing (Supplementary Fig. 8). Frozen sperm for each line is available from the authors upon request.

Adult zebrafish (*Danio rerio*) animals were maintained at 28 °C with standard 14/10 light/dark cycles. For dissections, we used adult Tubingen long-fin strain (~22 months old). Equal female and male zebrafish were euthanized in an ice bath of system water until the termination of buccal and gill motion. Tissue dissection was performed in physiological saline E3h media (5 mM NaCl, 0.17 mM KCl, 0.33 mM CaCl2, 0.33 mM MgSO4, and 1 mM HEPES, pH 7.3). Collected samples were immediately collected in centrifuge tubes and frozen.

**Clones, cell lines, and transfection**. A full length *Gnat1* clone (accession BC058810) was obtained from Horizon Discovery and recloned in pcDNA3.1(+) (Invitrogen). A matching clone in which all 16 TAG triplets in the 1.11kbp 3'-UTR were mutated to TGA to disrupt the Musashi binding sites was created using gene synthesis (Genscript). Gibson assembly was used to reclone the cDNAs into pcDNA3.1(+) vector and attach HA- and T7-tags to the wild type and mutant clone, respectively. Full length *Msi1* clone with N-terminal Flag epitope tag in pcDNA3.1 was described before[30]. Full length, codon optimized mouse *Pcbp2* clone with N-terminal Flag epitope tag was produced by gene synthesis (Genscript) and cloned in pcDNA3.1(+) (Invitrogen). All clones are available from Addgene (https://www.addgene.org/Peter_Stoilov/).

NIH 3T3 cells were grown in DMEM supplemented with 10% Fetal Bovine Serum. The cells were maintained in a humidified incubator at 37 °C in 5% CO$_2$ atmosphere. Transfection with polyethyleneimine was carried out as described before[57]. Briefly, the 6 h prior to transfection the cells were seeded at $3.2*10^5$ cells per well in 6 well plates. A total of 500 ng of DNA was used per transfection, containing 125 ng of each wild type and mutant *Gnat1* construct, and 250 ng of expression vector that carried a flag-tagged EGFP, *Pcbp2* or *Msi1* clone in pcDNA3.1(+) backbone. The cells were collected 27 to 29 h post-transfection to analyze protein and mRNA expression.

**RNA extraction and RT-PCR**. RNA was extracted with Trisol and precipitated with isopropanol. The RNA was then dissolved and treated with RNAse-free DNase I (Roche) for 20 min at 37 °C. After DNA digestion the reactions were extracted once with chloroform and the RNA was precipitated with ethanol. RT-PCR analysis of alternative splicing using fluorescently labeled primers was described before[30,58].

The levels of Gnat1 transcripts expressed from the recombinant clones in 3T3 cells were determined by multiplexed RT-qPCR. Hydrolysis probes to the HA and T7 tags were used to detect Gnat1 transcripts with wild type and mutant 3'-UTRs, respectively. The RT-qPCR was performed using Luna One Step RT-qPCR mix with dUDG (NEB). Amplification using Luna One Step qPCR mix with UDG (NEB) that did not contain a reverse transcriptase component (NEB) served as no-reverse-transcriptase controls for DNA contamination. The ratio of the transcript levels measured by the HA and T7 probes was used to determine the effect of each treatment on the mRNA levels expressed from the constructs carrying wild type and mutant 3'-UTRs. RT-qPCR was performed on Biorad CFX96 thermocycler with the parameters: RT reaction – 2 min at 25 °C, 10 min at 55 °C; Initial denaturation: 1 min at 95 °C; Amplification: 45 cycles of denaturation for 10 sec at 95 °C and annealing/extension for 30 sec at 60 °C. The primers and probes used for alternative splicing and RT-qPCR analysis are listed in Supplementary Data 9.

**Electroretinography (ERG) Measurement and Preparation of Animal**. ERGs were measured using either UTAS Visual Diagnostic System with Big-Shot Ganzfeld device (LKC Technologies, Gaithersburg, MD, USA) or Celeris system with Espion software (Diagnosis LLC, Lowell, MA, USA). Prior to testing, mice were dark-adapted overnight. All further handling of mice following dark adaptation was performed under deep red illumination. The mice were anesthetized by inhalation of 1.5% isoflurane mixed with 100% oxygen at a flow rate of 2.5 l/min. The pupils were topically dilated with a drop of tropicamide and phenylephrine-hydrochloride, allowing drops to sit on both eyes for 10 min. After that, mice were

transferred to a heated platform connected to a nose cone that allows for a continuous flow of isoflurane. A reference electrode was inserted sub-dermally between the eyes of the mouse, and ERG responses were collected from both eyes using wire electrodes placed on the center of each cornea, with contact being made using a drop of 0.3% Hypromellose solution. To deliver the stimulus, a Ganzfeld Bowel was used with LED white arrays at increasing intensities. Dark-adapted scotopic photoresponse was recorded under the dim red light using a single LED white light flashes of luminescence ranging from $2.45\cdot10^{-4}$ to 2.4 cd-s/m$^2$. For photopic response, animals were light-adapted for 10 min in the presence of rod-saturating 30 cd-s/m$^2$ ambient white light prior to recording the photopic response.

**Western Blot**. Mouse retinas were lysed using RIPA buffer (50 mM Tris HCl-pH 8.0, 150 mM NaCl, 1.0% TritonX-100, 0.5% sodium deoxycholate, 0.1% sodium dodecyl sulfate) supplied with protease (Sigma-Aldrich catalog# 535140-1 ML) and phosphatase inhibitors cocktail (Sigma-Aldrich catalog # P5726-1 ML). After homogenization, the lysate was incubated on ice for 10 min, then cleared by centrifugation for 15 min. 20 μg of protein extract was resolved in 4–20% polyacrylamide SDS–PAGE gel and transferred onto polyvinylidene difluoride (PVDF) membranes (Immunobilon-FL, Millipore). After blocking with BSA in PBST (Phosphate- buffered saline with 0.1% Tween-20), the membranes were blocked and probed with primary antibodies overnight at 4 °C, followed by incubation with fluorescently labeled (Alexa Fluor 647 or 488, Jackson ImmnuoResearch) secondary antibodies for 1 h at room temperature. The membranes were then scanned on Amersham Typhoon Phosphorimager (GE Healthcare). To quantify the protein expression across membranes the band intensities detected by each antibody were scaled to the median signal for the membrane. The scaled expression values were then normalized to the scaled values of the corresponding controls (loading controls, or in the transfection experiments to proteins expressed from co-transfected constructs). The antibodies used for western blot analysis are listed in Supplementary Data 11.

**Retinal histology**. The whole eyecups from the knockout and control mice were enucleated. The eyes were then fixed using a Z-fixative (Excalibur Pathology Inc). Tissue processing, including paraffin embedding and hematoxylin and eosin (H&E) staining was performed at Excalibur Pathology. Images of the stained slides were collected using a Nikon Brightfield Microscope operated by Element software (Nikon). To evaluate the photoreceptor cell loss, we counted the number of nuclei within the outer nuclear layer (ONL) using the NIS elements software. The counting was done at ten equidistant locations centered on the optical nerve and moving toward the periphery in 400 μm increments. Five locations were on the inferior side (−5 to −1) and five on the superior side (1–5) of the retina relative to the optic nerve. For each location and the number of nuclei reported is the average of four technical replicates. The nuclei counts were averaged over 3 biological replicates that represent retinas from three separate animals.

**Immunocytochemistry**. Eyes were enucleated, and small window was cut in the cornea before immersing it in 4% paraformaldehyde fixative (4% PFA in PBS: 137 mM NaCl, 2.7 mM KCl, 10 mM Na$_2$HPO$_4$, and 1.8 mM KH$_2$PO$_4$, pH 7.2) for 3 h on a rotator. Eyecups were dehydrated by sequential incubation in 7.5%, 15%, and 22% sucrose in 1xPBS. Eyecups were then snap-frozen in optimal cutting temperature compound (OCT) blocks. Serial 16 μm sections were cut on a Leica CM1850 cryostat and mounted onto Superfrost Plus microscope slides (Fischer Scientific). Mounted retinal sections were washed three times for 10 min each with PBS and then blocked with PBST for 1 h (10% goat serum, 0.3% TritonX-100, and 0.02% sodium azide in PBS). Retinal sections were incubated overnight at 4 °C with primary antibodies diluted in PBST supplemented with 5% goat serum. After three 15 min washes with PBST the sections were incubated for one hour with secondary antibodies diluted 1:1000 in PBST supplemented with 5% goat serum and 4',6-diamidino-2-phenylindole. The sections were washed three times for 15 min with PBST, mounted with Prolong Antifade reagent (Thermofisher) and secured with coverslips. The sections were imaged on a Nikon C2 laser scanning confocal microscope. The laser power, gain and offset settings were maintained the same when imaging sections from knockout and control littermates. The antibodies used for immunofluorescence staining are listed in Supplementary Data 11.

**RNA sequencing**. Total RNA was isolated at day 21 post-tamoxifen injection using Tri-reagent (Sigma) from retinas in four biological replicates of *Msi1*$^{+/+}$/*Msi2*$^{+/+}$ and *Msi1*$^{-/-}$/*Msi2*$^{-/-}$ mice. Sequencing libraries were prepared by the West Virginia University genomics core using KAPA Hyper RNA with Riboerase (Roche). The libraries were sequenced by the University of Illinois DNA services core on Illumina HiSeq 4000 at average depth of 44 million 100 nt paired end reads.

RNA-Seq reads were aligned to the mouse genome (GRCm38) using HISAT2[59]. The mapped reads were summarized using Rsubread and differential gene expression analysis was carried out by edgeR (Supplementary Data 1)[60,61]. Inclusion levels of cassette exons were calculated by rMATS (4.1.0), using reads spanning exon-exon junctions[62].

**CLIP-sequencing and meta exon analysis**. Retinas from wild-type mice (80 mg per replicate, two replicates) were collected and placed in ice-cold PBS drops on a 10 cm plate. Plates containing retinas were then placed on ice and UV-crosslinked (254 nm, 200 mJ/cm$^2$) using UV Stratinker$^{TM}$ 2400. UV-crosslinked retinas were then snap frozen in liquid nitrogen. Further tissue processing, eCLIP library prep, and sequencing were carried out by Eclipse Bioinnovations following a previously published protocol[63]. The rabbit anti-MSI1 (1:1000; catalog# ab 52865, Abcam, Cambridge, MA) was used for eCLIP. A total of four libraries were prepared—two from independent immunoprecipitations and two size matched sample inputs. Average library size was ~200 nt, corresponding to ~65 nt insert, excluding the 10 nt unique molecular identifier (UMI) barcode. The libraries were sequenced on Illumina HiSeq 4000 to a depth of between 101 and 118 million reads per library. The raw data obtained from the eCLIP-Seq are available at the NCBI Sequence Read Archive under project accession PRJNA795195.

To analyze the raw eCLIP data, the adapter sequences were first trimmed using Cutadapt[64]. HISAT2 was used to map the reads to version GRCm38 of the mouse genome and the mapped reads were deduplicated by umi-tools using the UMI barcodes built into the adapters[59,65]. Total of 24 million and 39 million unique reads from the input libraries, and 50 million and 58 million unique reads from the IP libraries were mapped to the genome. Crosslink sites were identified and clustered (Supplementary Data 19) into regions using PureClip[66]. Motifs enriched in the 51 nucleotide sequence fragments centered on the crosslink sites were identified by HOMER and DREME[67,68].

Meta-exon analysis was performed using the RBP-maps software package on non-redundant sets of alternatively spliced exons identified in our RNA-Seq analysis of *Msi1/Msi2* double knockout in photoreceptor cells[69]. The distribution of MSI1 crosslinks around exons downregulated or upregulated in the photoreceptor-specific *Msi1/Msi2* double knockout was compared to alternative exons that were not affected by the knockout (Supplementary Data 12, 13, and 14). 1000 random permutations of the non-regulated exon set were used to determine the 99.5% confidence intervals as described by Yee et al.[69].

**Proteomics analysis**. Retina samples from *Msi1$^{+/+}$/Msi2$^{+/+}$* and *Msi1$^{-/-}$/Msi2$^{-/-}$* were collected 21 days after tamoxifen injections. Five biological replicates were used for each wild-type and knockout group. Tissue processing and proteomics quantification of snapped frozen retina samples was performed by IDeA proteomics. Briefly, proteins were reduced, alkylated, and purified by chloroform/methanol extraction prior to digestion with sequencing grade modified porcine trypsin (Promega). Tryptic peptides were labeled using tandem mass tag isobaric labeling reagents (Thermo) following the manufacturer's instructions and combined into one 10-plex sample group. The labeled peptide multiplex was separated into 46 fractions on a 100 × 1.0 mm Acquity BEH C18 column (Waters) using an UltiMate 3000 UHPLC system (Thermo) with a 50 min gradient from 99:1 to 60:40 buffer A:B ratio under basic pH conditions, and then consolidated into 18 super-fractions. Each super-fraction was then further separated by reverse phase XSelect CSH C18 2.5 um resin (Waters) on an in-line 150 × 0.075 mm column using an UltiMate 3000 RSLCnano system (Thermo). Peptides were eluted using a 60 min gradient from 98:2 to 60:40 buffer A:B ratio. Eluted peptides were ionized by electrospray (2.2 kV) followed by mass spectrometric analysis on an Orbitrap Eclipse Tribrid mass spectrometer (Thermo) using multi-notch MS3 parameters. MS data were acquired using the FTMS analyzer in top-speed profile mode at a resolution of 120,000 over a range of 375 to 1500 m/z. Following CID activation with normalized collision energy of 35.0, MS/MS data were acquired using the ion trap analyzer in centroid mode and normal mass range. Using synchronous precursor selection, up to 10 MS/MS precursors were selected for HCD activation with normalized collision energy of 65.0, followed by acquisition of MS3 reporter ion data using the FTMS analyzer in profile mode at a resolution of 50,000 over a range of 100–500 m/z. Buffer A is 0.1% formic acid, 0.5% acetonitrile; Buffer B is 0.1% formic acid, 99.9% acetonitrile. Both buffers were adjusted to pH 10 with ammonium hydroxide for offline separation.

To create a database of proteins expressed in the retina we first filtered our mouse retina RNA-Seq data to remove genes with median expression across all samples that were below the median expression for the dataset. As a result, we selected 15,626 genes with expression equal or >1.2 RPKM. Ensembl release 79 was queried for annotated proteins produced by these genes resulting in a database of 34,055 protein sequences. Peptide identification against the retinal protein database was performed using MS-GF+ (version v2021.03.22) with parent ion tolerance of 10 ppm, reporter ion tolerance of −0.00335 Da and +0.0067 Da, and requiring fully tryptic peptides[70]. Only peptides with peptide level Q-value of 0.05 or below were accepted. The MSnbase package from R/Bioconductor was used to quantify the MS3 reporter ions and combine the identification and quantification data[71,72]. Differential protein expression analysis was performed using the DeqMS package from R/Bionconductor[73]. Protein changes with adjusted *p*-value below 0.05 and fold change of >1.5 were considered significant.

WebGestalt (WEB-based Gene SeT AnaLysis Toolkit) was used to perform enrichment analysis on the Gene Ontology and KEGG databases for the proteins and transcripts (Supplementary Data 6 and 7) with significant changes in gene expression. Only terms enriched at FDR < 0.05 were reported[74].

**Statistics and Reproducibility**. Age-matched males and females in the C57BL6/J background were used in all experiments. Analyses were performed on multiple

biological replicates as stated throughout the text. Biological replicates represent cell lines samples grown and transfected independently, different animals, and tissue samples collected from different animals. The statistical analysis and data visualization was done using GraphPad Prism and R/Bioconductor. Unpaired Student's *t*-test was used to assess statistical significance between control and knockout samples. Statistical significance was determined with one-way or two-way ANOVA followed by pairwise comparisons as indicated in the text. All data were presented as the mean ± standard error of the mean. RNASeq, eCLIP and Protemics experiments were performed on distinct sample sets. Sample sizes: eCLIP $n = 2$; RNA-Seq $n = 4$; mass-spectrometry $n = 5$; Quantification of outer nuclear layer thickness $n = 3$; ERG experiments $n \geq 3$; RT-PCR and Western Blot experiments $n \geq 3$.

**Reporting summary**. Further information on research design is available in the Nature Research Reporting Summary linked to this article.

## Data availability

The eCLIP-Seq and RNA-Seq data is available at the NCBI Sequence Read Archive under project accessions PRJNA795195 and PRJNA795137. The mass spectrometry proteomics are deposited to the ProteomeXchange Consortium via the PRIDE partner repository with the dataset identifier PXD030748 and 10.6019/PXD030748[75,76]. Source data underlying figures are provided in Supplementary Data 15 (Figs. 2, 4, 8 and Supplementary Figs. 2, 4, 6), Supplementary Data 16 (Fig. 6b), and Supplementary Data 18 (Fig. 7). Uncropped versions of blots and gels are provided as Supplementary Figs. 1, 3, 7, 9, 12 and 13.

## Code availability

R scripts and parameter files used in the RNASeq and mass-spectrometry analysis are deposited at GitHub (https://github.com/stoilov-lab/publications/tree/main/Matalkah_etal_ComBiol_2022). Uncropped gel images used in the preparation of Fig. 8 panels c and d are shown on supplementary Fig. 13. The original image files are available from the authors upon request.

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

## Acknowledgements

This work was supported by the National Institutes of Health 2R01EY025536 (P.S. and V.R.), Visual Sciences Center of Biomedical Research Excellence (VS-CoBRE grant P20GM144230), and bridge funding provided by the West Virginia University Health Sciences Center Office of Research and Graduate Education (P.S.). E.J.H is supported by West Virginia University and Department of Biology startup funds, Research and Scholarship Advancement award, and the Program to Stimulate Competitive Research. We are grateful to Dr. Christopher Lengner for the generous donation of the *Msi1*$^{fl/fl}$ and *Msi2*$^{fl/fl}$ mice and Dr. Steve Tsang for creERT2 animals. We thank Drs. Roberta Leonardi and Aaron Robart for critical reading of this manuscript.

## Author contributions

F.M. experimental conception, experimental design, data acquisition, data analysis, data interpretation, manuscript writing, and manuscript revision. B.J. data acquisition and data analysis. M.S. data acquisition. E.H. experimental design, provided zebrafish tissue samples. V.R. conception, experimental design, data interpretation, and manuscript revision. P.S. conception, experimental design, data acquisition, data analysis, data interpretation, and manuscript revision.

## Competing interests

The authors declare no competing interests.
