## [Peer Review File · Communications Biology]

Reviewers' comments:

Reviewer #1 (Remarks to the Author):

In this manuscript, the authors find that individual deletion of Msi1 and Msi2 are indistinguishable from control retinas, suggesting that each protein can compensate for loss of the other. The authors also demonstrate that while Msi1 and Msi2 are required for photoreceptor-specific splicing events, CRISPR/Cas9 deletion of photoreceptor-specific exons in *Ttc8*, *Cep290*, *Cc2d2a*, *Cacna2d4*, *Slc17a7*, and a combinatorial *Ttc8/Cep290/Cc2d2a* deletion results in normal retina morphology and responses to light. Finally, the authors present data that suggest Msi1 and Msi2 may play more important roles in regulating post-transcriptional expression of genes that are essential for photoreceptor survival.

Overall, the manuscript is well designed and clearly written. It is unfortunate that the photoreceptor-specific exon deletion experiments (Figure 6) did not result in retinal deficits, but such results are important to publish. As the authors indicate, it is unclear what selective pressures are acting on these exon sequences to preserve their photoreceptor specificity. However, given that *Ttc8* and *Cc2d2a* are absent in certain other species, it remains possible that individual deletion of other photoreceptor-specific exons could be important for retinal function. Perhaps future whole genome sequencing data may reveal retinal dystrophy mutations that are adjacent to photoreceptor-specific exons that would motivate such studies to be performed.

Specific comments:

1. In Figure 1, it should be indicated in the legend that Day 0 of tamoxifen injection is mouse age P30 (as mentioned in the main text).
2. If isotropic mass spec analysis of individual Msi1(-/-) or Msi2(-/-) were to be performed, would the authors expect the protein levels to be different from Msi1(+/+)/Msi2(+/+) controls? Is there a known dose-dependence for the effect of Msi1/Msi2 on downstream gene expression?
3. If Msi1 or Msi2 were to be delivered to the retina via photoreceptor minipromoter AAV, would the authors expect increased expression of differentially changed genes from Figure 7? Or do the authors believe that Msi1 and Msi2, given their high expression levels, are close to saturation in normal retina.

Reviewer #2 (Remarks to the Author):

In this manuscript, Matakah et al. investigate the role of Musashi proteins, Msi1 and Msi2, in the normal function of mature photoreceptors. Using a series of knockout mice, the authors show that the expression of Msi proteins is essential for mature photoreceptor function and survival, however, their roles are redundant as single Msi1 or Msi2 knockouts do not have a detectable phenotype. Applying eCLIP, authors identify Msi1 binding sites in the transcriptome of photoreceptor cells; although a significant fraction of binding sites are intronic, the majority are in the 3'-UTRs. Extensive characterization of the effects of Msi proteins reveals their role in inclusion of several conserved exons in photoreceptor specific transcripts, however, loss of these splicing events is found to not have any physiological consequence. Importantly, the data shows that Msi proteins regulate expression of photoreceptor specific genes predominantly through modulation of translation. Overall, the experiments are elegantly designed and meticulously performed, and manuscript is very well written. The following, mostly minor, changes are recommended.

1. Page 9, lines 13-20 – authors compare Msi1/Msi2 regulated exons to Aipl1 regulated exons, however, rationale or significance for this comparison is not provided. Some background on Aipl1 should be included.
2. Page 7, line 9 should be changed to “.....the single layer of photoreceptor cells is remaining in the ONL at day 113”.
3. Page 8, line 13 should say “.....enhanced UV crosslinking and”, as eCLIP is different from CLIP.

4. Supplemental Figure 1 and 3 are not mentioned anywhere on the Results or the Figure legends.
5. Page 14, paragraph 1 – a reference to Figure 8C that demonstrates expression of HA and T7 tagged Gnat1 proteins should be added.

Reviewer #3 (Remarks to the Author):

Mataalkah et al

The Musashi proteins activate post-transcriptionally protein expression and alternative exon splicing in vertebrate photoreceptors

In this study, the authors employed a conditional Msi1/Msi2 KO mouse targeting mature photoreceptor cells and NGS and MS approaches to characterize the role of the two Musashi isoforms in these terminally differentiated cells. The authors provide compelling data that nuclear Msi1/Msi2 function redundantly in the mature photoreceptors to promote splicing of photoreceptor-specific alternative exons. Interestingly, this Musashi-dependent alternative splicing appears dispensable for photoreceptor function, at least on the mRNAs tested. To address the cytoplasmic role of Msi1/Msi2, the authors employed RNAseq and MS on separate sample sets and argue that Msi1/2 predominantly act to promote translation of photoreceptor enriched mRNAs. This is a very interesting study and I support publication if the Gnat1 mRNA reporter assays can be corrected experimentally with some additional comments addressed in the text.

1. Gnat1 mRNA reporter assay (Figure 8C and 8D)

Figures 8C and 8D purport to show that the Gnat1 mRNA 3' UTR promotes Musashi-directed mRNA translational activation. The authors employ co-transfection of HA-tagged wild-type Gnat1 mRNA and 3' UTR or T7-tagged Gnat mRNA with a mutant 3' UTR where all TAG motifs were mutated to TGA to disrupt Musashi binding. These constructs were then mixed and co-transfected with empty vector or plasmids expressing GFP, or FLAG-tagged Msi1 or FLAG-tagged Pcbp2. Following western blot quantitation (Fig 8C), the relative expression of wild-type to mutant 3'UTR GNAT1 is presented as a ratio (Fig. 8D, HA/T7). Although the ratios suggest 2-fold activation of Gnat mRNA translation after co-expression of Musashi1 with only a minor increase in mRNA level (Fig 8D), this is misleading. Examination of the levels of HA-GNAT1 expression (Fig 8C) does not support a 2-fold increase as the levels of HA-GNAT1 are comparable to empty vector control (ie. no translational increase is observed relative to the empty vector). Moreover, levels of T7-tagged GNAT1 controlled by the mutant Gnat1 3' UTR actually show repression versus the empty vector control (about 50%), which accounts for the ~2 fold increase in the HA/T7 ratio. In other words, when referenced against the empty vector control the translational activation is an artifact of the HA/T7 ratio analysis. I am not convinced that an empty vector is the best control to use. I think expression of GFP serves as a better control in these experiments and if this is the case, relative to GFP there does appear to be ~2 fold increase in HA-GNAT1 with Msi1 expression (HA blot, Fig 8C) and no differences seen with the T7-GNAT. However, the GFP protein needs to be FLAG tagged to allow ectopic protein levels to be directly compared to that of the Pcbp2 and Msi1 constructs, hence the request to repeat these analyses. An RNA binding mutant Msi1 would be a nice optional specificity control.

2. The title is grammatically awkward and unjustifiably proposes an activation role for Musashi-directed translational control, when the authors show significant potential repression activity too. I would suggest a change to "The Musashi proteins direct post-transcriptional control of protein expression and alternate exon splicing in vertebrate photoreceptors".

3. The authors perform eCLIP and provide information of where Musashi binds on the recovered transcripts (eg 59.7% have Musashi bound to the 3' UTR). Clarify how many transcripts were recovered in total - I could not find this information.

4. Presumably Gnat1, Prom1 and Impg2 are eCLIP targets – if this was not explicitly stated it should be. This also goes for the other up or down regulated proteins (but not mRNAs) shown in Fig 8A (blue rombs and crosses) – these are all Musashi eCLIP targets?

5. IMPG2 should be shown in Figure 7

6. Related to point 4 above, of the 165 proteins differentially expressed between wild-type and Msi1/2 null photoreceptors, were all of the mRNAs encoding these Musashi eCLIP targets and how many of these 165 are blue in Fig 8A?

7. The authors should note in the text that the mass spectrometry and RNAseq analyses were performed on separately prepared sample sets. While there is a correlation between altered protein changes and a lack of significant mRNA transcript level changes between the analyses (Fig 8A), validation of these protein and mRNA observations has not been performed within the same sample set. This caveat should be noted.

8. On page 13 the authors discuss 2 sets of genes (photoreceptor enriched or genes with lower expression in photoreceptors. How many genes in each set? In Figure 8A, there is a trend for the photoreceptor specific proteins to have lower expression in the Msi1/2 mutants (except for PROM1 and IMPG2). The blue crosses in Fig 8A represent the second set of genes? Since Msi1/2 were deleted specifically in the photoreceptor cells, the proteins represented by the blue crosses were presumably also affected in the photoreceptor cells and do not represent changes in non-photoreceptor cell types – correct? The authors should provide numbers – how many blue rombs and how many blue crosses are we looking at in Fig 8A? What % of the 98 reduced protein expression or % of the 67 up regulated proteins do these represent. And clarify if all the blue proteins (rombs or crosses) are encoded by Musashi bound mRNAs from the eCLIP data. This is not linked very well in the text. Unless the authors can provide compelling evidence that the proteins represented by the blue crosses are changed in non-photoreceptor cells rather than in the photoreceptor cells, then it seems Musashi exerts BOTH translational repression and translational activation of target mRNAs in this system. As such I disagree with the conclusion statement on p13 that “Taken together our data demonstrates that the Musashi proteins promote protein expression post-transcriptionally”. Clearly, Musashi both promotes and represses protein expression, not just promotes translation. This needs to be corrected.

9. Related to point 8 above, it is very interesting that translation of “photoreceptor-specific” mRNAs appears to be predominantly activated by Musashi while the non-photoreceptor specific mRNAs appear to be predominantly repressed. The authors should speculate on what might distinguish the photoreceptor-specific mRNA 3’ UTRs to be biased towards activation of translation. Is there any obvious difference in TAG motif number or position in Prom1, Impg2 or candidate repressed 3’ UTRs from the Gnat1 3’ UTR or the other candidate activated mRNA 3’ UTRs?

10. The authors should include the length of the Gnat1 3’ UTR and how many TAGs were mutated in the Method section

11. The first paragraph of the Discussion can be deleted as it duplicates the Introduction

12. Reference 56 is a Biorxiv citation – what is journal policy on citing non-refereed material?

13. Page 19 – the authors state “In photoreceptors, the Musashi proteins act largely to promote protein expression.” I disagree (see point 8) Musashi both promotes and represses protein expression, not just promotes translation. This needs to be corrected.

14. Page 19 – I similarly disagree with the statement “Our study is the first to show a broad role for the Musashi proteins as almost exclusive post-transcriptional activators of protein expression.”

15. In Methods, please indicate the time after transfection when lysates were prepared for HA and T7 analyses in NIH3T3 cells

16. In Methods, please clarify the number of experimental replicates used. Also was a non-specific control Ab used? If not, what proportion of total mRNAs were recovered in the Msi eCLIP experiments?

17. In Figures 1A, 3A and 3B the MSI1 and MSI2 signals are found predominantly in a layer labeled IS. Please define IS and in the text explain briefly what this is and how it relates to the ONL

18. Out of curiosity, do the mRNAs that undergo MSI-dependent alternative splicing also show regulated mRNA translation (Fig 8A, activated or repressed) or are they mutually exclusive? Perhaps comment on this in the discussion

19. In Fig 8A, please label GNAT1

Response to reviewers

We appreciate the effort the reviewers made in evaluating our work and we thank them for the supportive and kind comments. Below are our responses to the specific comments and recommendations of each reviewer.

Reviewers' comments:

Reviewer #1 (Remarks to the Author):

In this manuscript, the authors find that individual deletion of Msi1 and Msi2 are indistinguishable from control retinas, suggesting that each protein can compensate for loss of the other. The authors also demonstrate that while Msi1 and Msi2 are required for photoreceptor-specific splicing events, CRISPR/Cas9 deletion of photoreceptor-specific exons in Ttc8, Cep290, Cc2d2a, Cacna2d4, Slc17a7, and a combinatorial Ttc8/Cep290/Cc2d2a deletion results in normal retina morphology and responses to light. Finally, the authors present data that suggest Msi1 and Msi2 may play more important roles in regulating post-transcriptional expression of genes that are essential for photoreceptor survival.

Overall, the manuscript is well designed and clearly written. It is unfortunate that the photoreceptor-specific exon deletion experiments (Figure 6) did not result in retinal deficits, but such results are important to publish. As the authors indicate, it is unclear what selective pressures are acting on these exon sequences to preserve their photoreceptor specificity. However, given that Ttc8 and Cc2d2a are absent in certain other species, it remains possible that individual deletion of other photoreceptor-specific exons could be important for retinal function. Perhaps future whole genome sequencing data may reveal retinal dystrophy mutations that are adjacent to photoreceptor-specific exons that would motivate such studies to be performed.

We thank the reviewer for the supportive comments. Regarding the potential for future genome sequencing to reveal retinal dystrophy mutations we would like to point our research into photoreceptor-specific alternative splicing was initiated by a one such mutation in the photoreceptor-specific exon of Ttc8 (Bbs8) that causes retinitis pigmentosa. The mutation did not produce insight into the function of the exon as it caused a frame shift that “knocked out” the protein in photoreceptor cells (Murphy et al Mol Cell Biol. 2015). Considering the lack of phenotype after deleting this exon in mice, the evidence we have so far points to the photoreceptor-specific exon in Ttc8 being a liability. This makes its conservation even more puzzling.

Specific comments:

1. In Figure 1, it should be indicated in the legend that Day 0 of tamoxifen injection is mouse age P30 (as mentioned in the main text).

Corrected as suggested. The legends on figures 1 (page 38 lines 823-824) and 3 (page 41 lines 8852-8853) now read: "... collected 14 days after tamoxifen injection at postnatal day 30 ..."

2. If isotropic mass spec analysis of individual Msi1(-/-) or Msi2(-/-) were to be performed, would the authors expect the protein levels to be different from Msi1(+/+)/Msi2(+/+) controls? Is there a known dose-dependence for the effect of Msi1/Msi2 on downstream gene expression?

We have not directly tested the effect of single Msi1 or Msi2 knockouts on protein expression. As the single knockouts did not show a phenotype, we reason that it is unlikely to see dramatic changes in protein expression in these animals. Furthermore, we consistently see moderate (1.3 to 1.5 fold) upregulation of MSI2 protein expression after knockout of *Msi1*. This was observed in our tamoxifen induced model and when in the past we used Six6-Cre to delete *Msi1* early in retinal development (Sundar et al JBC 2021). Because of the apparent regulation Msi2 by Msi1 the interpretation of the gene dosage experiments would not be straightforward. Nevertheless, we cannot rule out gene dosage effects on protein expression that do not result in observable phenotype.

3. If Msi1 or Msi2 were to be delivered to the retina via photoreceptor minipromoter AAV, would the authors expect increased expression of differentially changed genes from Figure 7? Or do the authors believe that Msi1 and Msi2, given their high expression levels, are close to saturation in normal retina.

We have not performed these experiments yet. We hope that Musashi overexpression could boost protein expression in cases of autosomal dominant Retinitis Pigmentosa where the phenotype is driven by reduced gene dosage. However, Musashi overexpression may only have an effect on low affinity targets. We think that the high affinity Musashi targets in the cytoplasm are close to saturation because photoreceptors cells accumulate significant amounts of MSI1 and MSI2 proteins in the nucleus (Murphy et al, PLoS genetics 2016). Such nuclear accumulation would only be possible if the two proteins saturate their cytoplasmic targets, as binding to RNA blocks the nuclear translocation of Musashi (Kawahara et al, JBC, 2008). The reason is that the nuclear localization signals of Musashi are located on the RNA binding surfaces of RRM1 and RRM2 (Kawahara et al, JBC, 6 May 2011) and NLS residues required for nuclear import are also involved in binding to RNA (Ohyama et al, NAR, 2012; Iwaoka et al, Molecules, 2017). Consequently only Musashi proteins that are not associated with RNA in the cytoplasm can be imported into the nucleus.

Reviewer #2 (Remarks to the Author):

In this manuscript, Matakah et al. investigate the role of Musahi proteins, Msi1 and Msi2, in the normal function of mature photoreceptors. Using a series of knockout mice, the authors show that the expression of Msi proteins is essential for mature photoreceptor function and survival, however, their roles are redundant as single Msi1 or Msi2 knockouts do not have a detectable phenotype. Applying eCLIP, authors identify Msi1 binding sites in the transcriptome of

photoreceptor cells; although a significant fraction of binding sites are intronic, the majority are in the 3'-UTRs. Extensive characterization of the effects of Msi proteins reveals their role in inclusion of several conserved exons in photoreceptor specific transcripts, however, loss of these splicing events is found to not have any physiological consequence. Importantly, the data shows that Msi proteins regulate expression of photoreceptor specific genes predominantly through modulation of translation. Overall, the experiments are elegantly designed and meticulously performed, and manuscript is very well written. The following, mostly minor, changes are recommended.

Saying that “the experiments are elegantly designed and meticulously performed, and manuscript is very well written” is by far the highest praise anyone could hope to receive for their research. We are truly grateful for this comment.

1. Page 9, lines 13-20 – authors compare Msi1/Msi2 regulated exons to Aipl1 regulated exons, however, rationale or significance for this comparison is not provided. Some background on Aipl1 should be included.

Aipl1 does not regulate splicing. It is a photoreceptor-specific chaperone involved in the processing of farnesylated proteins. Animals lacking Aipl1 rapidly lose their photoreceptors. In Aipl1 knockout mice the photoreceptors are completely lost by postnatal day 30 while the rest of the retina is largely intact. In the past we used the Aipl1 knockout animals as a tool to identify transcripts and splicing variants that are specific to or expressed at higher level in photoreceptor cells (Murphy et al, PLoS Genetics, 2016). These splicing variants and transcripts were identified by comparing the transcriptome of a wild type retina were the photoreceptor cells comprise approximately 50% of the cells to that of the Aipl1 knockout retina which has no photoreceptors. We have changed the structure of this sentence so that it cannot be interpreted as to say that AIPL1 controls splicing. It now reads: “*Previously we identified a photoreceptor-specific alternative splicing program by comparing the splicing in wild type retina to that in retina that is devoid of photoreceptor cells due to knockout of the Aipl1 gene.*” (page 9, lines 166-168)

2. Page 7, line 9 should be changed to “.....the single layer of photoreceptor cells is remaining in the ONL at day 113”.

The meaning of this sentence is that photoreceptor cells are lost and only one layer, “a single layer”, of photoreceptor cells is left by day 113. Using a definite article “the” instead of indefinite “a” would mean that what is left is a specific layer of photoreceptor cells. This is not what we want to convey. To improve the clarity we changed the sentence to: “... *and only one layer of photoreceptor cells remained at day 113...*” (page 7, line 124)

3. Page 8, line 13 should say “.....enhanced UV crosslinking and”, as eCLIP is different from CLIP.

We agree. Corrected as suggested (page 8, line150)

4. Supplemental Figure 1 and 3 are not mentioned anywhere on the Results or the Figure legends.

These figures show the full size blots from which the images on Figure 1 and Figure 3 are derived. We added references to them in the figure legends of Figure 1 (page 38, lines 829) and 3 (page 41, lines 858-859) that state “*See Supplement Figure ... for full size blots.*” We have also updated the legends of supplementary figures 1 (supplementary figures, page 1 lines 1-5) and 3 (supplementary figures, page 4, lines 18-20).

5. Page 14, paragraph 1 – a reference to Figure 8C that demonstrates expression of HA and T7 tagged Gnat1 proteins should be added.

Corrected as suggested: “*...clones were distinguished by the HA (wild type) and T7 (mutant) epitope tags (Figure 8C and D).*” (page 14, lines 284-285).

Reviewer #3 (Remarks to the Author):

Matakah et al

The Musashi proteins activate post-transcriptionally protein expression and alternative exon splicing in vertebrate photoreceptors

In this study, the authors employed a conditional Msi1/Msi2 KO mouse targeting mature photoreceptor cells and NGS and MS approaches to characterize the role of the two Musashi isoforms in these terminally differentiated cells. The authors provide compelling data that nuclear Msi1/Msi2 function redundantly in the mature photoreceptors to promote splicing of photoreceptor-specific alternative exons. Interestingly, this Musashi-dependent alternative splicing appears dispensable for photoreceptor function, at least on the mRNAs tested. To address the cytoplasmic role of Msi1/Msi2, the authors employed RNAseq and MS on separate sample sets and argue that Msi1/2 predominantly act to promote translation of photoreceptor enriched mRNAs. This is a very interesting study and I support publication if the Gnat1 mRNA reporter assays can be corrected experimentally with some additional comments addressed in the text.

We thank the reviewer for the supportive evaluation and the meticulous review. We have performed the Gnat1 reporter assays as the reviewer suggested. We hope our answers and the changes to the text address the remaining issues the reviewer identified.

1. Gnat1 mRNA reporter assay (Figure 8C and 8D)

Figures 8C and 8D purport to show that the Gnat1 mRNA 3' UTR promotes Musashi-directed mRNA translational activation. The authors employ co-transfection of HA-tagged wild-type

Gnat1 mRNA and 3' UTR or T7-tagged Gnat mRNA with a mutant 3' UTR where all TAG motifs were mutated to TGA to disrupt Musashi binding. These constructs were then mixed and co-transfected with empty vector or plasmids expressing GFP, or FLAG-tagged Msi1 or FLAG-tagged Pcbp2. Following western blot quantitation (Fig 8C), the relative expression of wild-type to mutant 3'UTR GNAT1 is presented as a ratio (Fig. 8D, HA/T7). Although the ratios suggest 2-fold activation of Gnat mRNA translation after co-expression of Musashi1 with only a minor increase in mRNA level (Fig 8D), this is misleading. Examination of the levels of HA-GNAT1 expression (Fig 8C) does not support a 2-fold increase as the levels of HA-GNAT1 are comparable to empty vector control (ie. no translational increase is observed relative to the empty vector). Moreover, levels of T7-tagged GNAT1 controlled by the mutant Gnat1 3' UTR actually show repression versus the empty vector control (about 50%), which accounts for the ~2 fold increase in the HA/T7 ratio. In other words, when referenced against the empty vector control the translational activation is an artifact of the HA/T7 ratio analysis. I am not convinced that an empty vector is the best control to use. I think expression of GFP serves as a better control in these experiments and if this is the case, relative to GFP there does appear to be ~2 fold increase in HA-GNAT1 with Msi1 expression (HA blot, Fig 8C) and no differences seen with the T7-GNAT. However, the GFP protein needs to be FLAG tagged to allow ectopic protein levels to be directly compared to that of the Pcbp2 and Msi1 constructs, hence the request to repeat these analyses. An RNA binding mutant Msi1 would be a nice optional specificity control.

We agree with the reviewer that the empty vector is not an appropriate control for this experiment. As suggested we use flag-tagged GFP as control. MSI1 protein retains its ability to enhance expression of the wild type Gnat1 construct by approximately 2 fold when compared to the new flag-tagged GFP control. Figure 8 with updated panels C and D is on page 50 of the updated manuscript along with updated legend (page 51, lines 931 and also see below). Supplementary Figure 10 was added with images of the full sized blots from this experiment. To reflect the change in the experimental design we updated:

1. The results section (page14, lines 289-296).
2. Subsection "Clones, cell lines, and transfection" of the Materials and methods section (pages 21, lines 463-467)
3. Deleted the sentence "*GFP protein in transfection samples was detected by its intrinsic fluorescence in the 488nm channel*" from the subsection on western blotting (page 24, lines 512-513).

2. The title is grammatically awkward and unjustifiably proposes an activation role for Musashi-directed translational control, when the authors show significant potential repression activity too. I would suggest a change to “The Musashi proteins direct post-transcriptional control of protein expression and alternate exon splicing in vertebrate photoreceptors”.

We thank the reviewer for the suggestion. The title is now updated as suggested: “The Musashi proteins direct post-transcriptional control of protein expression and alternate exon splicing in vertebrate photoreceptors” page1, line1)

3. The authors perform eCLIP and provide information of where Musashi binds on the recovered transcripts (eg 59.7% have Musashi bound to the 3’ UTR). Clarify how many transcripts were recovered in total - I could not find this information.

In the RNA-Seq data we detect transcripts from 30,283 transcripts. Of these 10,161 have at least one MSI1 eCLIP peak and 7,849 have eCLIP peaks in the 3’-UTR. We have added a sentence to the results section with these numbers: “Out of 30,283 transcripts detected by RNA-Seq, 10,161 had at least one eCLIP peak enriched over input and 7,849 transcripts had eCLIP peaks in the 3’-UTR (Supplementary Table 8)” (page 8, lines 156-158). Supplementary Table 8 is updated with a column indicating which transcripts have eCLIP peaks in their 3’-UTRs.

4. Presumably Gnat1, Prom1 and Impg2 are eCLIP targets – if this was not explicitly stated it should be. This also goes for the other up or down regulated proteins (but not mRNAs) shown in Fig 8A (blue rombs and crosses) – these are all Musashi eCLIP targets?

This is correct, Gnat1, Prom1 and Impg2 are eCLIP targets. All 39 regulated proteins (but not mRNA) that are derived from “photoreceptor specific” genes (blue rombs) contained MSI1 eCLIP peaks in their UTRs. Their non-photoreceptor specific counterparts (blue crosses) were

29 in total of which 17 contained MSI1 eCLIP peaks. Having said that, the presence of eCLIP in non-photoreceptor specific targets is irrelevant in the context of an experiment where Musashi is deleted only on photoreceptor cells. The results section is updated to state: “All 39 proteins with altered expression derived from “photoreceptor-specific” genes (blue rombs on Figure 8A) contained MSI1 eCLIP peaks in their 3'-UTRs. (page 13 lines 267 - 268). Supplementary table 7 is updated with a column indicating which transcripts have eCLIP peaks in their 3'-UTRs.

5. IMPG2 should be shown in Figure 7

We added IMPG2 in place of Cc2d2a on the leftmost panel on row 3 of Figure 7 (page 48 and the image below). We reformatted the figure to contain two combined sections for “Outer segment and Cilium” proteins and for “Intraflagellar transport and BBSome” proteins. The changes are reflected in the figure legend (page 49, lines 912-915).

6. Related to point 4 above, of the 165 proteins differentially expressed between wild-type and Msi1/2 null photoreceptors, were all of the mRNAs encoding these Musashi eCLIP targets and how many of these 165 are blue in Fig 8A?

The 165 differentially expressed proteins were detected in whole retina samples. Of these, 121 have MSI1 eCLIP peaks in the 3'-UTRs. Of the 165 differentially expressed proteins 80 are plotted on Figure 8A. The remaining proteins were excluded from the dataset because they are expressed at comparable levels between photoreceptor and non-photoreceptor cells. These proteins were excluded from our analysis because it would have been hard to deconvolute to what degree changes in their expression are derived from photoreceptor cells vs other cells in the retina.

7. The authors should note in the text that the mass spectrometry and RNAseq analyses were performed on separately prepared sample sets. While there is a correlation between altered protein changes and a lack of significant mRNA transcript level changes between the analyses (Fig 8A), validation of these protein and mRNA observations has not been performed within the same sample set. This caveat should be noted.

As suggested we stated this caveat in the discussion: *“The amount of material required for RNASeq, mass-spectrometry, and eCLIP-Seq experiments did not allow us to perform the experiments in parallel on samples from the same animal. Samples from different animals were used for each experiment. While there is an excellent correlation across replicates, it should be noted that changes in protein and RNA expression were not determined within the same animal.”* (page 18 lines 388-392). We also updated the Statistics and Reproducibility subsection in the Material and Methods section: *RNASeq, eCLIP and Proteomics experiments were performed on distinct sample sets.”* (page 29, line 630).

8. On page 13 the authors discuss 2 sets of genes (photoreceptor enriched or genes with lower expression in photoreceptors. How many genes in each set? In Figure 8A, there is a trend for the photoreceptor specific proteins to have lower expression in the Msi1/2 mutants (except for PROM1 and IMPG2). The blue crosses in Fig 8A represent the second set of genes? Since Msi1/2 were deleted specifically in the photoreceptor cells, the proteins represented by the blue crosses were presumably also affected in the photoreceptor cells and do not represent changes in non-photoreceptor cell types – correct? The authors should provide numbers – how many blue rombs and how many blue crosses are we looking at in Fig 8A? What % of the 98 reduced protein expression or % of the 67 up regulated proteins do these represent. And clarify if all the blue proteins (rombs or crosses) are encoded by Musashi bound mRNAs from the eCLIP data. This is not linked very well in the text. Unless the authors can provide compelling evidence that the proteins represented by the blue crosses are changed in non-photoreceptor cells rather than in the photoreceptor cells, then it seems Musashi exerts BOTH translational repression and translational activation of target mRNAs in this system. As such I disagree with the conclusion statement on p13 that “Taken together our data demonstrates that the Musashi proteins promote protein expression post-transcriptionally”. Clearly, Musashi both promotes and represses protein expression, not just promotes translation. This needs to be corrected.

This is incorrect. The genes labeled with crosses, including the “blue crosses”, were specifically selected because they are not expressed in photoreceptor cells, or are expressed at levels that are at least two fold lower in photoreceptor cells compared to any other cell type in the retina. Thus changes in their expression in photoreceptor cells are unlikely to significantly affect the overall levels of these proteins in the retina. As discussed in the manuscript the “blue crosses” tend to be proteins expressed in glial cells. The change in their expression does not reflect direct regulation by Musashi. We think these genes represent the response of glia and perhaps other retinal cells to photoreceptor dysfunction. As described in the answers to questions above, the numbers of genes in each category are now stated in the results section. Supplementary tables, 4, 7 and 8 are updated to indicate which genes have 3'-UTRs with eCLIP peaks.

9. Related to point 8 above, it is very interesting that translation of “photoreceptor-specific” mRNAs appears to be predominantly activated by Musashi while the non-photoreceptor specific mRNAs appear to be predominantly repressed. The authors should speculate on what might distinguish the photoreceptor-specific mRNA 3' UTRs to be biased towards activation of translation. Is there any obvious difference in TAG motif number or position in Prom1, Impg2 or candidate repressed 3' UTRs from the Gnat1 3' UTR or the other candidate activated mRNA 3' UTRs?

We are reluctant to speculate because this is something we are actively working on. We do not see any obvious differences in Musashi binding site position or binding site density. The small number of the high confidence regulated “photoreceptor-specific” proteins (39 in total) and the lack of a high confidence baseline set of transcripts underpower our attempts to look for enriched motifs and common features. We have indications that interaction with the RNA interference machinery may be involved in the regulation of protein expression by Musashi. We just obtained eCLIP data for AGO2 from wild type and Musashi knockout retina samples and are in the process of analyzing it.

10. The authors should include the length of the Gnat1 3' UTR and how many TAGs were mutated in the Method section

The material and method section is updated with the number of TAGs mutated (16 TAGs were mutated) and the length of the 3'-UTR (1.11kbp): “A matching clone in which all 16 TAG triplets in the 1.11kbp 3'-UTR were mutated to TGA to disrupt the Musashi binding sites was created using gene synthesis...” (page 21 lines 452-453). The clones with complete sequences are now available from Addgene (https://www.addgene.org/Peter_Stoilov/).

11. The first paragraph of the Discussion can be deleted as it duplicates the Introduction

Deleted as suggested (page 15 lines 304-315).

12. Reference 56 is a Biorxiv citation – what is journal policy on citing non-refereed material?

We had two references citing preprints in Biorxiv: Karmakar, S. et al, and Clampi et al. Both manuscripts are now published in peer-reviewed journals and we have updated the citations accordingly (references 56 and 45) .

13. Page 19 – the authors state “In photoreceptors, the Musashi proteins act largely to promote protein expression.” I disagree (see point 8) Musashi both promotes and represses protein expression, not just promotes translation. This needs to be corrected.

We disagree with the reviewer regarding point 8. Among the proteins with significant change in expression that is not dictated by the corresponding mRNA levels we can confidently claim that 39 are coming from photoreceptor cells. Of these 37 are downregulated upon Msi1/Msi2 knockout. In contrast most of the proteins that are upregulated in the knockout originate from non-photoreceptor cells. In addition, the cumulative frequency plot on Figure 8B shows that there is a general decrease in protein levels of “photoreceptor-specific” genes that is not matched by changes in mRNA levels.

14. Page 19 – I similarly disagree with the statement “Our study is the first to show a broad role for the Musashi proteins as almost exclusive post-transcriptional activators of protein expression.”

We have removed this statement (page 19, lines 406-408). We agree that it is not well supported.

15. In Methods, please indicate the time after transfection when lysates were prepared for HA and T7 analyses in NIH3T3 cells

The collection time was 27-29 hours after transfection. This is now stated in the materials and methods section: “*The cells were collected 27 to 29 hours post-transfection to analyze protein and mRNA expression.*” (page 22, lines 466-467).

16. In Methods, please clarify the number of experimental replicates used. Also was a non-specific control Ab used? If not, what proportion of total mRNAs were recovered in the Msi eCLIP experiments?

We apologize for the omission. We have updated the article text and figure legends with the number of replicates (legends for figures 7 and 8 on pages 41 and 58 were missing this information). Where possible the replicates are shown as individual points on the plots. We also provide the data underlying the figures as supplementary tables that contain the values of the individual replicates.

Regarding the eCLIP experiment, we used two replicates. The eCLIP protocol does not involve use of non-specific antibody control. Instead, the IP signal in eCLIP is normalized to size-matched input. For the IP we used a MS11 monoclonal antibody that we have shown to be specific by mouse knockout (Sundar et al, JBC 2021; and this manuscript). The specificity of the

IP is also confirmed by subsequent analysis that shows the top motifs enriched in the eCLIP to contain the canonical UAG binding site for the Musashi proteins. The crosslink sites are in close proximity to the UAG binding site indicating direct binding.

The eCLIP protocol does not provide for quantifying RNA recovery in the IP as the RNA fragments are ligated to adapters on the IP beads. Typically only a small fraction of RNA is crosslinked and recovered in the IP, so much so that in the original CLIP protocol the cross-linked RNA was tracked by radiolabeling. The doses of UV required to achieve complete crosslinking of all binding sites will also damage the RNA and impede the subsequent ligation and reverse transcription steps of the protocol.

17. In Figures 1A, 3A and 3B the MSI1 and MSI2 signals are found predominantly in a layer labeled IS. Please define IS and in the text explain briefly what this is and how it relates to the ONL

We added a sentence to the results to indicate that the Inner Segment (IS) and the Outer Nuclear Layer (ONL) are composed of photoreceptor cells and represent roughly the cytoplasm and the nuclei of the photoreceptors: *“The immunofluorescence signal is lost from both the cytoplasm (inner segment, IS) and the nuclei (outer nuclear layer, ONL) of the photoreceptor cells”* (page 6, lines 97-99).

18. Out of curiosity, do the mRNAs that undergo MSI-dependent alternative splicing also show regulated mRNA translation (Fig 8A, activated or repressed) or are they mutually exclusive? Perhaps comment on this in the discussion

Some RNAs undergo both MSI dependent alternative splicing and protein expression regulation (Prom1, Ttc8), while others are regulated only on pre-mRNA splicing level (Cc2d2a, Atp1b2) or only translationally. We do not see evidence for coordination between the two processes, which is expected considering they are carried out in different cellular compartments. We have updated the discussion to address this question: *“As the Musashi proteins are regulating both pre-mRNA splicing in the nucleus and protein expression in the cytoplasm there are some genes affected by both modes of regulation, e.g. Prom1, raising the question of potential coordination between pre-mRNA splicing and protein expression. We do not see evidence for such coordination in our data”* (page 19, lines 403-406).

19. In Fig 8A, please label GNAT1

Labeled as suggested (see above for the updated figure 8A). While updating the figure we noticed an error in the plot presented on the original figure 8A: a data point was missing (approximate coordinates $x = 0$, $y = -1.8$) and two more data points were covered by the legend. We re-plotted the data and moved the legend to the upper left corner (page 50). The changes do not affect the conclusion we made based on the presented data.

REVIEWERS' COMMENTS:

Reviewer #1 (Remarks to the Author):

The authors have suitably addressed all my questions and concerns.

Reviewer #2 (Remarks to the Author):

The authors have satisfactorily addressed all of reviewer's concerns.

Reviewer #3 (Remarks to the Author):

Matakah et al have responded thoroughly to my prior critiques and have significantly bolstered the manuscript. The Gnat1 3' UTR reporter assay is now internally consistent and appropriately controlled. My other comments have all been satisfactorily addressed - the authors are to be commended for their efforts. I recommend acceptance of the manuscript.